



# 1  Channel evolution processes in a diamictic glacier foreland. Implications on downstream sediment supply: case study Pasterze / Austria

Michael Paster[1], Peter Flödl[1], Anton Neureiter[2], Gernot Weyss[2], Bernhard Hynek[2], Ulrich Pulg[3],
Rannveig Ø. Skoglund[4], Helmut Habersack[1], Christoph Hauer[1]
[1]CD-Laboratory for Sediment research and management, Institute of Hydraulic Engineering and River Research, Department
of Water, Atmosphere and Environment, University of Natural Resources and Life Sciences Vienna, Muthgasse 107, 1190
Vienna, Austria
[2]Department for Climate Research, Central Institute for Meteorology and Geodynamics, Hohe Warte 38, 1190 Vienna, Austria
[3]UNI Research Miljø, Laboratorium for Freshwater Ecology and Inland Fisheries, Nygårdsgaten 112, 5006 Bergen, Norway
[4]University of Bergen, Fosswinckelsgt. 6, 5006, Bergen, Norway
*Correspondence to*: *Michael Paster (michael.paster@boku.ac.at)*
**Abstract.** Global warming and glacier retreat are affecting the morphodynamics of proglacial rivers. In response to changing
hydrology, their altered hydraulics will significantly impact future glacifluvial erosion and proglacial channel development.
This study analysis proglacial channel evolution processes at the foreland of Austria's biggest glacier Pasterze by predicted
runoff until 2050. A high-resolution digital elevation model was created by an unmanned aerial vehicle, channel bathymetry
was sampled, a one-dimensional hydrodynamic-numerical model was generated, and bedload transport formulas were used to
calculate the predicted transport capacity of the proglacial river. Due to the fine sediment composition near the glacier terminus
($d_{50}$< 49.6 mm), the calculation results underline the process of headward erosion in the still unaffected, recently deglaciated
river section. In contrast, an armor layer is already partly established by the coarse grain size distribution in the already incised
river section ($d_{50}$> 201 mm). Furthermore, already reoccurring exposed non-fluvial grain sizes combined with decreasing flow
competence in the long term are indicators for erosion-resistant pavement layer formation and landform decoupling in the
vertical direction. The presented study clearly shows that subsystems of 'developed channels' exhibiting pavement formation
of non-fluvial deposits are found at the investigated glacier foreland. Thus, an extension accompanied by a refinement of the
fluvial system in the sediment cascade approach was developed as a central result.

## 25  1 Introduction

Since the Little Ice Age (LIA) around 1850, global warming has caused temporal and spatial changes in high mountain areas
by glacier retreat (e.g., Zemp et al., 2019; Fischer et al., 2018; Huss et al., 2008) and permafrost decline (Harris et al., 2009).
While deglaciation of European glaciers has accelerated and repeatedly reached peak values in recent years (Sommer et al.,
2020), formerly glaciated areas are continuously expanding and are characterized by high geomorphological activity (e.g.,
Avian et al., 2018; Lane et al., 2016; Carrivick et al., 2013; Cavalli et al., 2013; Old et al., 2005; Gruber et al., 2004).



Deglaciated areas in direct proximity to the glacier terminus are termed proglacial (Slaymaker, 2009) and are confined by LIA
moraines (Heckmann & Morche, 2019). Within this steadily increasing spatial boundary, the amount of loose and
unconsolidated sediment exceeds the 'geological norm' defined by non-glaciated catchments. Proglacial areas are, therefore,
transitional landscapes that adapt to this geological norm within the paraglacial period (Ballantyne, 2002; Church & Ryder,
1972). This adjustment occurs by various geomorphological processes (e.g., gully erosion, avalanches, debris flows), where
sediment is reworked along the gravitational gradient (Ballantyne, 2002). In contrast, continuous sediment supply is given by
(sub)glacial erosion (e.g., Hallet et al., 1996; Alley et al., 2019) and moderately well-rounded (Benn & Evans, 2013) poorly
sorted unconsolidated material ranging in size from sand to cobbles up to boulders (diamictic till; Harland et al., 1966) is
deposited in the outwash plain by meltwater (Benn & Evans, 2013). The sediment production and reworking process chain of
(temporary) sediment storages within a catchment can be described by the 'sediment cascade approach' (Chorley & Kennedy,
1971). The sediment connectivity between these storage landforms in longitudinal (in-stream linkage), lateral (e.g., channel –
hillslope relationship), and vertical (channel bed – subsurface connection) direction (Fryris et al., 2007) is highly dynamic
(Lane et al., 2016) and crucial if sediment from different origins reaches the valley floor and contributes to the glacifluvial
transport in the proglacial channel network (e.g., Beylich et al., 2019; Brierley et al., 2006). Fluvial sediment evacuation is
considered as the last transport process of the sediment cascade (e.g., Geilhausen et al., 2012b; Etzelmüller & Frauenfelder,
2009; Schrott et al., 2003; Chorley & Kennedy, 1971) and is predominant in the paraglacial period (Church & Ryder, 1972).
Alpine proglacial areas are in general highly dynamic fluvial systems (e.g., Leggat et al., 2015; Micheletti et al., 2015; Mao et
al., 2014; Gurnell, 1995; Warburton, 1992), triggered by daily to seasonal meltwater fluctuations and high-magnitude/low-
frequency events (e.g., Baewert & Morche, 2014; Marren, 2005; Beylich & Gintz, 2004). Combined with the high sediment
supply by erosion of glacifluvial diamictic till, braided channels emerge in direct glacier proximity (e.g., Maizels, 2002;
Gurnell et al., 1999; Ashworth & Ferguson, 1986). More downstream with increasing distance to the glacier terminus,
depending on (i) sediment composition, (ii) runoff variability, (iii) channel slope, and (iv) potential confinement (e.g.,
moraines, glacier ice), the channel turns into a single thread river (e.g., Marren, 2005; Gurnell et al., 1999). Another dominant
process supporting the formation of single channels, however, is river bed incision when the transport capacity exceeds the
sediment supply (e.g., Wilkie & Clague, 2009; Marren, 2005; Gurnell et al., 1999). This kind of glacifluvial process leads to
the exposure of non-fluvial sediment in formerly glaciated environments and creates an armor layer (Bunte & Abt, 2001).
Fluvial sediment transport, which mainly contributes to the stabilization of proglacial areas, especially with increasing distance
to the glacier terminus (e.g., Delaney et al., 2018; Lane et al., 2016), is described in the sediment cascade as a glacifluvial
process (Geilhausen et al., 2012b). Whether this process is able for sediment transport or sediment remains deposited is defined
by the hydraulic parameter 'flow competence' – defined as the largest particle a flow can move (Benn & Evans, 2013).
Flow competence is mainly impacted by the runoff conditions, which are predicted to change by global warming (e.g., Förster
et al., 2015; Farinotti et al., 2012; Braun et al., 2000). The glacier mass of the Austrian Alps is expected to decrease
continuously (Fischer et al., 2018), which implies changes in the future glacial discharge regimes: (i) on a short time scale,
glacial meltwater will increase due to deglaciation, (ii) in a long-term perspective, the runoff will decrease by 'exceeding the





expected moment of peak water' (Schaefli, 2015; Farinotti et al., 2012). This exceedance is predicted before 2050 for European
glaciers (Huss & Hock, 2018); after that, the runoff will lose its glacial characteristic over time. Alongside these predictions,
(i) the annual peak runoff will be shifted to spring (Förster et al., 2015), and (ii) reduced average peak runoff will hence the
bedload transport of proglacial rivers (Pralong et al., 2015). All these predictions mainly affect the flow competence of rivers
and impact channel bed stabilization by glacifluvial erosion, the last process of proglacial sediment cascade models.
This study aims to predict the effect of global warming on proglacial channel evolution. For this purpose, the proglacial part
of the river Möll at the foreland of Austria's biggest glacier Pasterze was investigated. Currently, the sediment yield of the
Pasterze catchment consists mainly of suspended sediment (Avian et al., 2018; Geilhausen et al., 2012b). Whether this behavior
remains the same in the future by changing runoff characteristics was investigated using predicted runoff by 2050. A high-
resolution digital elevation model (DEM) was created for hydrodynamic numerical modeling, and bedload transport formulas
were used to predict the proglacial channel's flow competence. The ongoing establishment of a pavement layer by (exposed)
non-fluvial sediment in sections with greater distance to the glacier forces a landform decoupling. The results obtained allow
a revision and extension of the fluvial system of the sediment cascade approach by incorporating the effects of global warming.

## 2 Study site

The investigation area is located in Carinthia in the national park Hohe Tauern at the foreland of the Pasterze Glacier
(47°5'8" N; 12°42'27" E), the biggest glacier in Austria and the Eastern Alps (16.2 km² in 2012). The glacier tongue (~4 km
length) is characterized by (i) a high mean annual rate of retreat of up to -50 m a$^{-1}$ (Fischer et al., 2018) and (ii) a debris coverage
of around 75 % (Kellerer-Pirklbauer, 2008). The total length loss of the Pasterze Glacier since LIA amounts to -2200 m until
2015 (Fischer et al., 2018). The debris mantle at the southern part (orographic right) of the glacier tongue results in a lower
deglaciation rate of up to 35 % by a minimum debris thickness of 15 cm (Kellerer-Pirklbauer et al., 2008). The proximal glacier
foreland is characterized by a low gradient (Geilhausen et al., 2012b; Krainer & Poscher, 1992), debris-covered dead ice
landforms (e.g., Avian et al., 2018; Seier et al., 2017; Geilhausen et al., 2012a) and one main proglacial river.

### 2.1 Proglacial river

The investigated reach covers around 850 m between the glacier terminus (2100 m a.s.l.) and the inflow (delta area) into the
continuously increasing lake 'Pasterzensee' (upstream of the lake "Sandersee", which formed in the late 1950s; Krainer &
Poscher, 1992) at 2070 m a.s.l. (Fig. 1). The channel is composed of four distinct sections: (i) the flat headwater near the
glacier terminus ($L$= 200 m; $S_m$= 1.3 %), (ii) a transition section ($L$= 100 m; $S_m$= 2.9 %) into (iii) the canyon (L= 482 m; $S_m$=
6.8 %), and (iv) the flat outlet into the delta area ($L$= 60 m; $S_m$= 1.7 %) of the lake 'Pasterzensee'. Almost the entire investigated
proglacial channel (except the delta area) is confined by the debris-covered glacier tongue and debris-covered dead ice (with
slower melting rates). The runoff behavior shows typical glacial characteristics with (i) high summer (up to $Q_{max}$= 25 m³s$^{-1}$)
and low winter runoff (down to $Q_{min}$= 0.1 m³s$^{-1}$) and (ii) strong seasonal and diurnal fluctuations (Krainer & Poscher, 1992).

**Figure 1:** Location of the study site: **(a.)** Carinthia, Austria; **(b.)** proximal foreland of the Pasterze Glacier, where the dashed rectangle indicates the proglacial river Möll including *(1)* glacier tongue (clean), *(2)* glacier tongue (debris-covered), *(3)* Pasterzensee, *(4)* Sandersee, *(5)* Kaiser-Franz-Josefs-Höhe; **(c.)** study reach, based on the own UAV survey, supplemented by the measuring sites for sediment analysis.

## 2.2 Sediment budget

The proximal foreland of the Pasterze Glacier is characterized by glacifluvial deposits, including big boulders, gravel, and sand (Fig. 2), which partly cover dead ice landforms (e.g., Avian et al., 2018; Seier et al., 2017; Geilhausen et al., 2012a). This moderately well-rounded, poorly sorted outwash (glacial diamictic till; Harland et al., 1966) is decoupled from the active




hillslopes around the proximal foreland, resulting in a transport-limited glacifluvial transport system (Geilhausen et al., 2012b).
The potential for paraglacial reworking on the overall sediment output is low compared to glacifluvial processes, especially
with increasing distance to the glacier terminus (Geilhausen et al., 2012a; 2012b). However, this proglacial area is still a
dynamic system with a high potential for fluvial reworking processes (Avian et al., 2018). The biggest proportion of the
sediment output is assumed as suspended load (Geilhausen et al., 2012b).

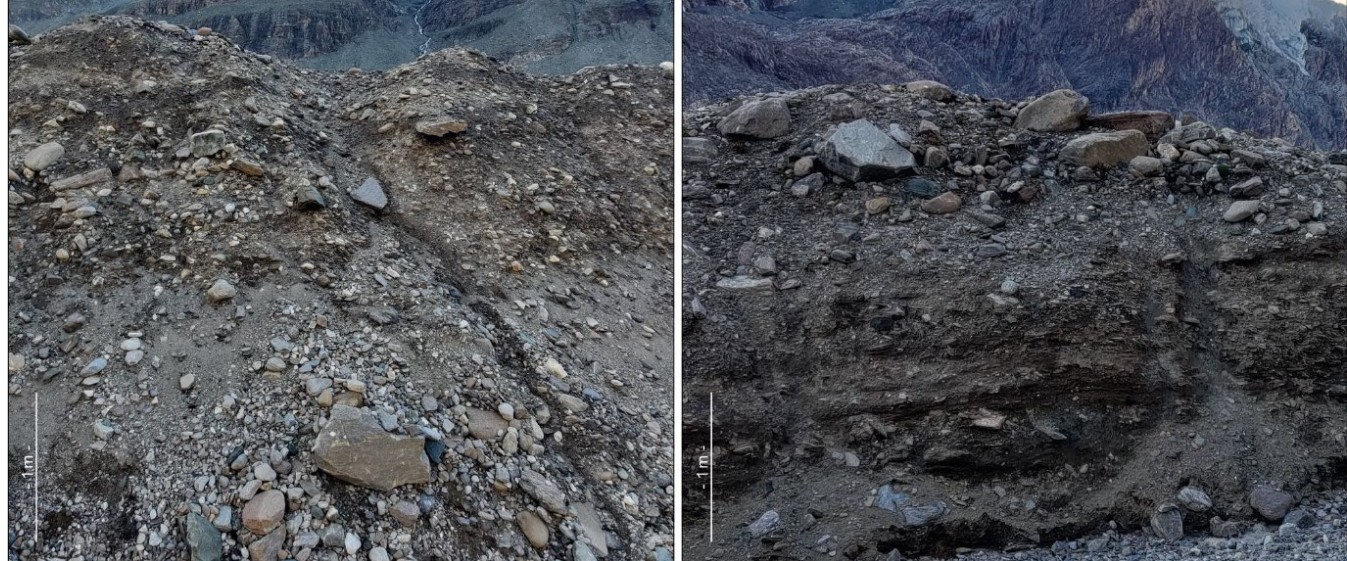

**Figure 2:** River embankment of the investigated proglacial canyon already incised in the poorly sorted diamictic outwash plain (photographs
taken during fieldwork).

## 3 Methods

### 3.1 UAV survey

The mapping was carried out during low flow conditions in autumn 2018, where the 850 m long river stretch (Fig. 1) was
covered by an unmanned aerial vehicle (UAV; type: hexacopter KR 615) equipped with a compact camera (type: Sony ILCE-
6000; focus length 16 mm) mounted on a stabilized gimbal. The survey was performed in two stages: (i) the entire study area
was covered with a constant flight level of 55 m above the river bed and (ii) the canyon in a second flight with a constant flight
level of 20 m above the channel bed (approx. surrounding terrain level). In total, 1371 photos (6000x4000 px) were taken,
whereby a requested overlap of 80 % (forward) and 60 % (sideward) was achieved. Before the flights, ground control points
(GCPs) were placed along the banklines to improve the geodetic accuracy of the digital elevation model (DEM). Due to limited
accessibility and high and steep channel embankments, no GCPs were laid out in the channel. All GCPs were mapped by an
RTK-enabled GNSS device (type: Emlid Reach RS2).



### 3.2 DEM preparation

In post-processing, the software PhotoScan by Agisoft was applied to create (i) a 3D point cloud and (ii) an orthomosaic according to the principle of Structure-from-Motion (SfM). This approach uses images taken from multiple perspectives to compute a 3D surface based on image-matching algorithms combined with multi-view stereo techniques (MSV). This process allows the calculation of the camera position and orientation (Snavely et al., 2008). The mapped GCPs were used for geo-referencing the model and accuracy assessment of the transformation (Fonstad et al., 2013).

First, 1371 photos were used in the alignment, the camera position and the orientation of the individual photos were estimated, and a sparse point cloud with 365 080 points was calculated. The mapped coordinates of 10 GCPs were assigned for geo-referencing in the next step. The sparse point cloud was purged to (i) remove high outliers and misaligned points (down to 295 371 points), (ii) optimize the camera position, and (iii) minimize the error between the GCPs. This refinement, including the accuracy assessment by the remaining four GCPs, led to a root-mean-square error (RMSE) of 0.056 m ($X_{RMSE}$= 0.025 m; $Y_{RMSE}$= 0.044 m; $Z_{RMSE}$= 0.024 m). In the third step, the DEM was calculated (478 231 187 points; 3940 points m$^{-2}$) with a ground sample distance (GSD) of 1.59 cm px$^{-1}$, and an orthomosaic was arranged.

### 3.3 Sediment sampling

The sediment sampling was done by the commonly used method for gravel to cobble-bed mountain rivers according to Fehr (1987). For the accessible river sections, the line-by-number method (LbN) was applied, where all grains (b-axis) along the projection of a line are sampled and measured (at least 150 stones). Four LbN-analyses at characteristic points were carried out and mapped with the RTK GNSS device (circles in Fig. 1). For the inaccessible canyon, sediment analysis was photogrammetrically in post-processing on the images taken during the UAV mapping. At six characteristic points (triangles in Fig. 1), the grains were measured manually according to the on-site method by Fehr (1987). Both applied methods only take the coarse fractions into account (partially grain size distribution), which was sufficiently accurate for the objectives of this study: while Fehr (1987) suggests the cut-off at $b \geq 1$ cm, the truncation for adequate identification of grains in the digital method is strongly dependent on the GSD ranging between $b > 10$-15 px (Detert et al., 2018) and $b > 4$ px (Lang et al., 2021).

### 3.4 Hydraulic Modeling

A one-dimensional hydrodynamic-numerical (HN) model was set up (using the software Hec-Ras by the United States Army Corps of Engineers) for calculating the hydraulic parameters (i) bed shear stress and (ii) energy gradient, both relevant for the used bedload transport formulas. For this objective, cross-sections (CS) at a 10 m maximum distance were generated from the high-resolution DEM. The point density was reduced (down to 490 points per CS) by applying the automatic point filter algorithm with minimum area change. The modeling was performed with steady runoff conditions and the predicted maximum mean monthly runoff until 2050 ($Q_{m.max}$), which was determined by the 'Glacier Runoff Evolution Model (GERM)' (Schöner




et al., 2013). The inaccessibility necessitated a sensitivity analysis for roughness determination by varying representative
roughness values (Strickler coefficient $k_{st}$), which resulted in $k_{st}$= 28 m$^{1/3}$s$^{-1}$ (headwater and delta) and $k_{st}$= 20 m$^{1/3}$s$^{-1}$ (canyon).

### 3.5 Initiation of motion

The calculation for the initiation of motion was done by the bedload transport formula for steep mountain channels according
to Eq. (1) by Rickenmann (1990). To consider the increased flow resistance due to large roughness elements in the canyon,
the energy gradient was reduced according to Eq. (2) by Rickenmann et al. (2006).
$$q_c = 0.065 * \left(\frac{\rho_s}{\rho_w} - 1\right)^{1.67} * g^{0.5} * I_R^{-1.12} * d_{50}^{1.5} \qquad (1)$$
Here, the specific discharge $(q_c)$ is a function of the characteristic grain diameter $(d_{50})$, the energy gradient $(I_R)$, and the ratio
between sediment $(\rho_s)$ and fluid density $(\rho_w)$. The calculation results according to this 'conventional approach' in this study
are termed $d_{50.c}$.
$$I_{red} = I_R * \left[\frac{n_r}{n_{tot}}\right]^a \qquad (2)$$
Here, the reduced energy gradient $(I_{red})$ is calculated by the ratio of the grain roughness $(n_r)$ and total roughness $(n_{tot})$, $a$= 1.5
is a constant. The calculation results with the reduced energy gradient are labeled with $d_{50.r}$ in this study.
The characteristic grain size $d_{90}$, required for this calculation step, was derived from the adjusted Wolman count method for
the entire canyon, as Hauer & Pulg (2018) described. According to this field-based method, the assumed b-axis of the three
largest stones were manually measured in each cross-section of the canyon on the high-resolution UAV aerial images. In total,
159 stones were measured ($b$= 546-3715 mm), which resulted in a mean $d_{90}$= 1290 mm for the entire canyon.

### 3.6 Determination of morphological changes

Due to the lack of multitemporal terrain data, a comparative analysis based on an orthophoto of 2015 was performed to
reconstruct the evolution of the proglacial channel (formed in the ablation season of 2015). The channel formation was verified
by continuously recorded images from an automatic camera installed at the Kaiser-Franz-Josefs-Höhe (Fig. 1). Due to the
recording rate of 5 minutes, the proglacial area can be observed in a high temporal resolution (compare Avian et al., 2020).

## 4 Results

### 4.1 Sediment analysis

The results for all ten grain size distribution curves can be described as narrowly graded, reflected in a very steep gradient of
each curve (Fig. 3a). The sediment composition in the direction of flow is becoming increasingly coarse $(d_{50.m:LbN.1}$= 29.9 mm
$< d_{50.m:LbN.2}$= 49.6 mm $< d_{50.m:LbN.3}$= 79.6 mm) with the same distribution in the delta area $(d_{50.m:.LbN.4}$= 40.3 mm) as in the
headwater. The evaluation of the UAV-based sediment measurements (six characteristic points in the canyon; triangles in Fig.





1) illustrate a much coarser composition $(d_{50.m:CS500}= 202.5$ mm; $d_{50.m:CS408}= 219.1$ mm; $d_{50.m:CS327}= 241.2$ mm; $d_{50.m:CS252}=$
201.3 mm; $d_{50.m:CS168}= 211$ mm; $d_{50.m:CS52}= 116.3$ mm). Large particles were measured in every characteristic point (up to $d_{90.m}$
$= 850$ mm), and the largest grain size was detected in the steepest part of the entire proglacial channel ($b= 3700$ mm).

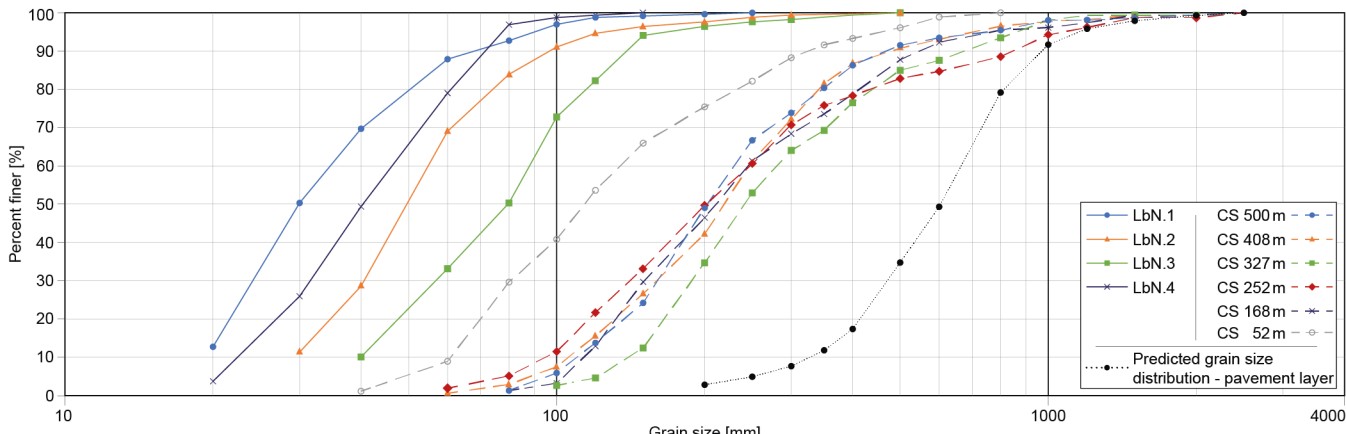

**Figure 3:** Partial grain size distribution curves: **(a.)** four line-by-number (LbN) analyses (continuous lines) and photogrammetric evaluations
(dashed lines) for six characteristic points in the inaccessible canyon. **(b.)** the dotted black curve refers to the potential future grain size
distribution of the pavement layer.

## 4.2 Development of flow competence

The seasonal course of the flow competence (the largest particle a flow can move) runs parallel to the typical glacial discharge
regime: smaller transportable grain sizes in the cold months and largest transportable grain sizes in the ablation season (summer
months). According to the forecasted hydrograph, the maximum mean monthly runoff $(Q_{m.max})$ will continuously increase in
the ablation seasons until June 2030 ($Q_{m.max.2030}= 14.61$ m³s⁻¹), following a decrease until 2050 ($Q_{m.max.2015}= 12.74$ m³s⁻¹), which
will be again around the level of 2018 ($Q_{m.max.2018}= 12.19$ m³s⁻¹). Crucial for this runoff development may be the exceedance
of the expected moment of peak water after 2030, where the maximum mean monthly meltwater runoff $(Q_{m.melt.max})$ is predicted
to decrease by two orders of magnitude until 2050 ($Q_{m.melt.max.2030}= 7.03$ m³s⁻¹ $>>$ $Q_{m.melt.max.2050}= 3.50$ m³s⁻¹).
A detailed consideration of the calculated flow competence (characteristic grain sizes $d_{50.c}$; $d_{50.r}$) according to $Q_{m.2030}$ and the
grain size measurements $(d_{50.m})$ in the longitudinal course shows two contrary results between (i) the flat headwater and (ii)
the canyon. The maximum calculated characteristic grain sizes near the glacier terminus (CS 842 m – CS 600 m; $S_m$: 1.3 %;
no big roughness elements; Fig. 1) by the conventional approach according to Rickenmann (1990) are bigger than those
determined on-site by the LbN-analysis (up to $d_{50.c}= 59.4$ mm $>$ $d_{50.m}= 49.6$ mm; Fig. 4a). In the transition section (CS 650 m
– CS 550 m) with a slightly increased channel gradient ($S_m= 2.9$ %), a much bigger characteristic grain size was calculated
then measured ($d_{50.c}= 275.5$ mm $>$ $d_{50.m}= 79.6$ mm). For the flow competence in the steep canyon (CS 550 m – CS 62 m; $S_m=$
6.8 %; big roughness elements), the opposite was observed, as the calculated characteristic grain size with the reduced energy
gradient $(I_{red})$ is smaller than measured in the aerial images (up to $d_{50.r}= 220$ mm $<$ $d_{50.m}= 241.2$ mm; Fig. 4b). The beginning
of the canyon is defined by the steepest part of the entire proglacial channel (around CS 512 m; $S_{max}= 18.9$ %), where a so-



called knickpoint developed resulting in the largest calculated characteristic grain sizes ($d_{50.r}$= 320 mm). The calculation results
indicate for all characteristic points in the canyon that the measured characteristic grain sizes $(d_{50.m})$ theoretically exceed the
calculated flow competence $(d_{50.r})$ by order of 1.1-1.6 at the maximum predicted discharge $(Q_{m.max.2030})$ in June 2030 (Fig. 4).

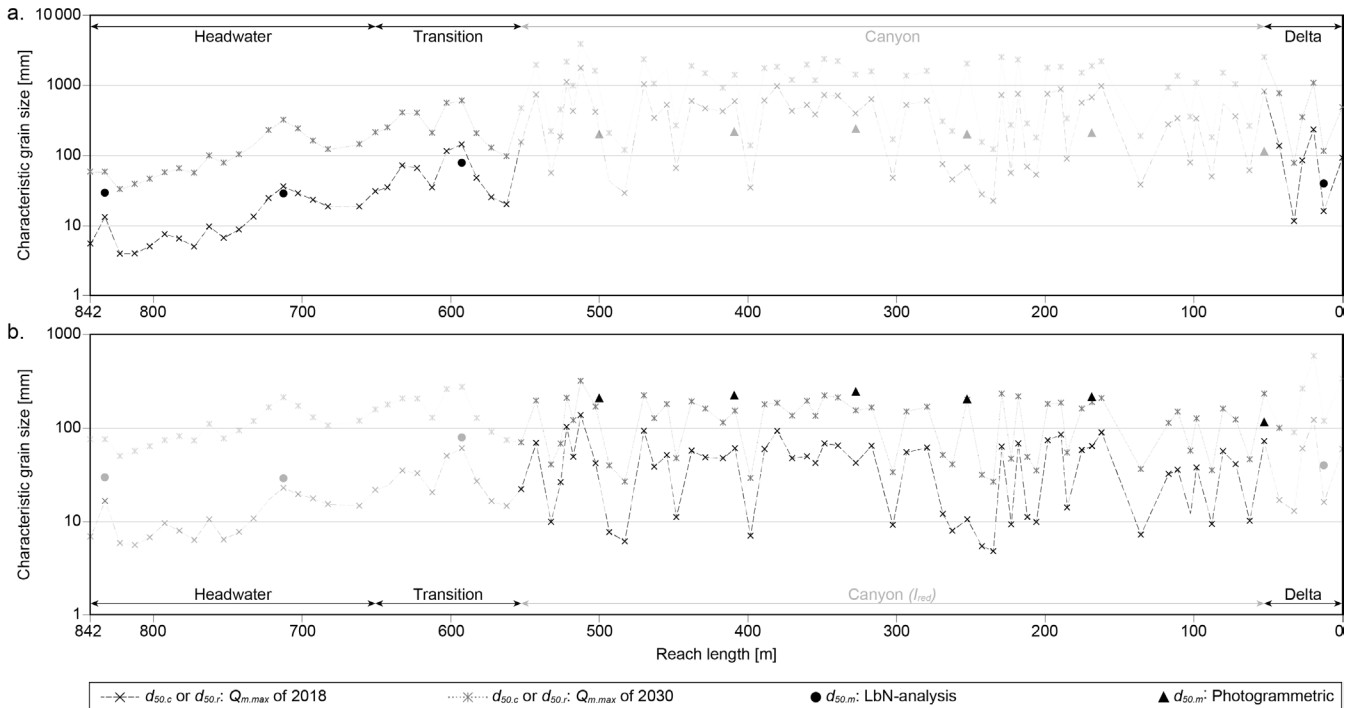


**Figure 4:** Longitudinal course of the calculated characteristic grain sizes (flow competence) according to **(a.)** Rickenmann (1990) $(d_{50.c})$ and
**(b.)** Rickenmann et al. (2006) with the reduced energy gradient $(d_{50.r})$. The transparently displayed parts of the graphs are outside the scope
of the respective approach and invalid for the respective sections. Each graph is supplemented by the measured characteristic grain sizes
$(d_{50.m})$ on-site (circle) and those evaluated photogrammetrically in post-processing (triangle). The results refer to the maximum predicted
mean monthly runoff by 2050 in June 2030 $(Q_{m.max.2030})$ compared to 2018 $(Q_{m.max.2018})$.
**4.3 Past morphological alterations**
The comparison with the orthophoto of 2015 (the most recent orthophoto before the fieldwork started) shows the steepest part
in 2018 (around CS 512 m, $S_{max}$= 18.9 %) at the glacier terminus in 2015. The onset of the canyon formation in the ablation
season of 2015 was verified by images from the automatic camera (Fig. 5a) installed at the 'Kaiser-Franz-Josefs-Höhe' (Fig.
1). While no fluvial channel is visible in the ablation season of 2014, a pronounced river structure can be detected one year
later (August 2015; Fig. 5a). The automatically recorded images indicate a very stable canyon and a highly dynamic delta area
since its development. The channel pattern in this fluvial deposition zone can change annually between braided (in 2016) and
single thread (in 2017). Due to the confinement by the debris-covered dead ice landforms and their slower melting rate, the
lateral changes $(\Delta B)$ in the canyon remained largely constant except (i) at the beginning of the canyon ($B$= +20 m) and (ii) in
the most downstream part ($B$= +15 m; Fig. 5b) as this part was still glaciated in 2015.







**Figure 5:** Proglacial fluvial channel: **(a.)** annual development stages between 2014 and 2019 recorded by the automatic camera (images provided by Großglockner Hochalpenstraße); **(b.)** comparison of flow paths after the channel formation in 2015 (orthophoto) and in the orthomosaic of 2018 (created from the own UAV survey).





## 5 Discussion

### 5.1 Channel evolution process

Channel bed incision as a stabilization process (e.g., Wilkie & Clague, 2009; Marren, 2005; Gurnell et al., 1999) was confirmed in this study, but with a remarkable longitudinal differentiation. Separated by a knickpoint (e.g., Hilgendorf et al., 2020; Schlunegger & Schneider, 2005), the headwater in direct glacier proximity is transitioning to the incised canyon (Fig. 6). This knickpoint is defined by the highest gradient ($S_{max}$=18.9 %) of the entire investigated proglacial reach established by the glacier terminus in 2015 (Fig. 5). The analysis of the river bathymetry and the results by the hydrodynamic model show potential for river bed incision in the headwater and tendencies for stabilization processes in the canyon (Fig. 6).

Moreover, the shift and alteration of the runoff will cause limitations in the bedload transport (e.g., Pralong et al., 2015; Geilhauesen et al., 2012b) and channel stabilization tendencies by glacifluvial sediment reworking are given with increasing distance to the glacier terminus (e.g., Delaney et al., 2018; Lane et al., 2016; Gurnell et al., 1999). The dominant process in the headwater is headward erosion, already known from, e.g., a fluvial drainage basin in Switzerland (Schlunegger & Schneider, 2005). Starting from this point with the highest gradient *($S_{max}$)*, the glacifluvial erosion will shift the knickpoint more upstream (Hilgendorf et al., 2020). The first indicators of this development were detected up to 140 m upstream of the knickpoint (CS 512 m) in the transition section (CS 650 m – CS 550 m; Fig. 1), defined (i) by a much bigger flow competence (largest particle a flow can move) than in the headwater and (ii) the exposure of already very big non-fluvial sediments ($b$> 2000 mm). Similar to the canyon, fine fractions are expected to be transported continuously out of the headwater, which will result in progressive armoring of the channel bed by sediment coarsening (Bunte & Abt, 2001; Dietrich et al., 1989). Exactly this post-glacial fluvial development is already occurring in the steep canyon. The local sorting of the diamicton by glacifluvial erosion resulted in channel bed incision (Fig. 6). The calculation results, according to the approach with $I_{red}$ (Rickenmann et al., 2006), valid for torrential flow characteristics (e.g., Pralong et al., 2015; Nitsche et al., 2011), indicate armor layer formation in the entire canyon. Due to decreasing flow competence by changing hydrology in the long-term perspective (e.g., Huss & Hock, 2018; Förster et al., 2015; Schöner et al., 2013; Haeberli et al., 2011), the channel bed of the already incised canyon will stabilize at $d_{50}$= 600 mm from a hydraulic point of view (Fig. 3b). Less bedload transport at the foreland of the Pasterze was already observed by Avian et al. (2018) and Geilhausen et al. (2012b).

The progressive armoring by (i) glacifluvial erosion combined with (ii) decreasing flow competence in the long-term perspective (Pralong et al., 2015) will establish an erosion-resistant pavement layer. In contrast to the infrequently mobile armoring layer (Bunte & Abt, 2001), this development will prevent channel bed incision by exposing non-fluvial deposits of the diamictic glacier foreland (outwash plain). The beginning of this trend was already observed in characteristic points in the canyon (triangles in Fig. 1), where very coarse (non-fluvial) sediment composition was occasionally measurable (up to $d_{90}$= 850 mm; Fig. 3). These points indicate the assumption of limited channel bed incision in the future and are labeled 'erosion breakpoints' (Fig. 6). For rivers characterized by such post-glacial non-fluvial sediment, Hauer & Pulg (2018) implemented the term glacial-till cascade, which contributes remarkably to channel stabilization.



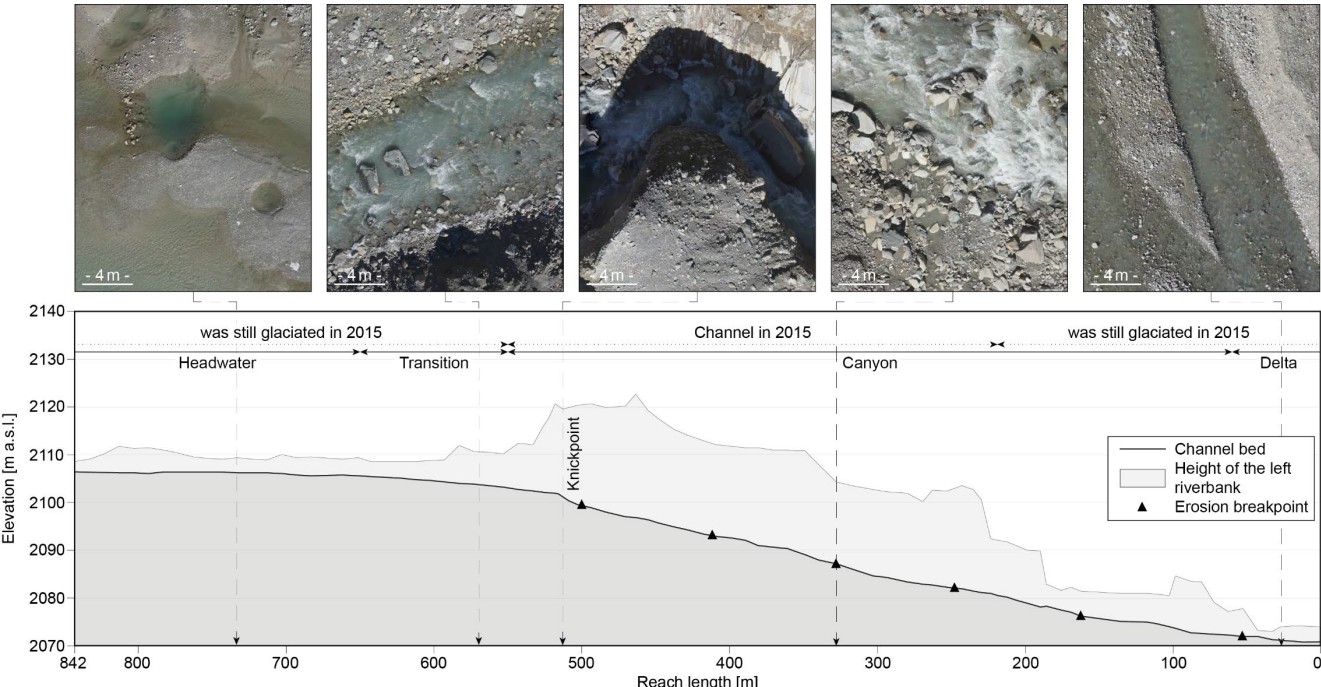

**Figure 6:** Longitudinal section of the investigated reach length with (i) the upper edge of the riverbank and (ii) predicted 'erosion breakpoints' according to the study results. The information about the glacierized area in 2015 originated from the orthophoto of 2015 (compare Fig. 5). In addition, pictures taken during UAV mapping show the sediment composition in some characteristic points.

The gradual evolution of proglacial channels in diamictic glacifluvial deposits starts as braided channels in direct glacier proximity (e.g., Marren, 2005; Gurnell et al., 1999). According to this development, glacifluvial erosion, predominant for sediment reworking (Church & Ryder, 1972), is defined in a generalized way as the last transport process of the proglacial sediment cascade model (Geilhausen et al., 2012b). However, according to the predicted study results, limited flow competence by 2050 (also compare Pralong et al., 2015) will develop a new in-stream storage type within the fluvial system of the sediment cascade model, defined by the erosion-resistant pavement layer (grey highlighted in Fig. 7). This extension by the new in-stream storage type (non-fluvial deposit) is accompanied by the refinement of the fluvial system within subsystem IV of the sediment cascade approach (dotted frames in Fig. 7). Glacifluvial erosion and the exposure of non-fluvial sediment lead inventible to a needed differentiation of the transport limited braided channel in direct glacier proximity and the (partly) supply limited 'developed channel' with a greater distance to the glacier terminus. Furthermore, established pavement layers disconnect the linkage between the proglacial channel bed and the unconsolidated diamictic sediment in the subsurface (e.g., Fryris et al., 2007; Brierley et al., 2006).





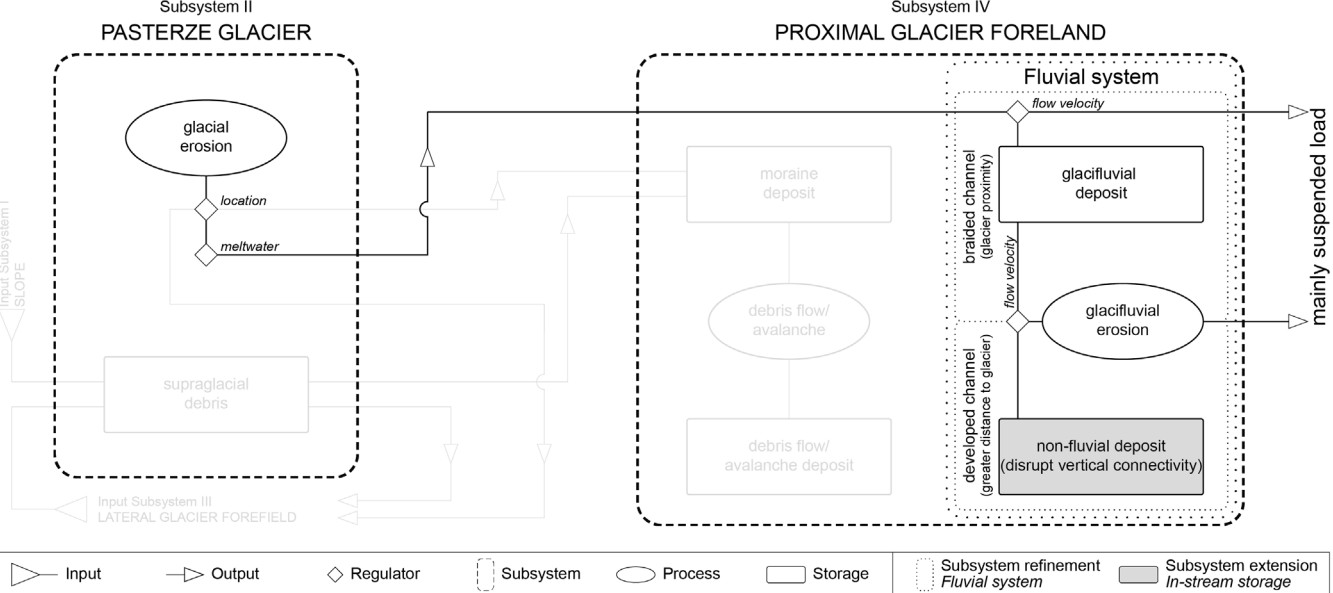

| Input | Output | Regulator | Subsystem | Process | Storage | Subsystem refinement *Fluvial system* | Subsystem extension *In-stream storage* |

**Figure 7:** Refinement (dotted frames) and extension (grey highlighted in-stream sediment storage type) by the fluvial system within subsystem IV (proximal glacier foreland) of the conceptual model of a sediment cascade for proglacial catchments. The establishment of a pavement layer by non-fluvial deposits will disrupt the vertical connectivity between the proglacial channel bed and the subsurface. Due to the decoupled subsystems in the catchment (Geilhausen et al., 2012b), the grey-coloured connections only complete the cascade model but are irrelevant to the objectives of this study. Modified after Geilhausen et al. (2012b).

Pavement layer formation by glacifluvial erosion is thus an essential stabilization as part of the well-known landform decoupling (e.g., Bakker et al., 2018; Wohl et al., 2015; Fryris et al., 2007) or (ii) vegetation cover within a river system (e.g., Eichel et al., 2018; Klaar et al., 2015; Gurnell et al., 1999). As the sediment cascade model shows decoupled subsystems in the Pasterze catchment (Geilhausen et al., 2012b), the new in-stream storage type is composed of non-fluvial glacial deposits. In order that the subsystems of a sediment cascade model are coupled with each other, coarse colluvial deposits can also be contained in this non-fluvial sediment storage type and contribute to channel stabilization (Carrivick & Rushmer, 2009). These developed channels will prevent further channel bed incision but will still allow lateral sediment supply, often triggered by high-magnitude/low-frequency events (e.g., Baewert & Morche, 2014; Marren, 2005; Old et al., 2005; Beylich & Gintz, 2004). Measurements in high mountain areas are prone to uncertainties, as (i) inaccessibility and (ii) torrential flow characteristics lead to limitations in the (i) geometry and calibration data acquisition as well as in sediment sampling. Due to low flow conditions during the in-situ measurements, representative sediment analysis of the canyon could be done in the partly wetted area. It is assumed that the same grain size composition is present in the permanently wetted area, although it will probably be already coarser due to constant exposition to glacifluvial erosive processes. However, the applied method of investigating the sediment composition (photogrammetrically in the inaccessible canyon) is satisfactory, as all grain sizes smaller than the lower threshold ($b > 65$ mm) are irrelevant for the pavement layer formation. Furthermore, simplifications were applied to the calculation approach (e.g., neglect of near-bed turbulence).



## 5.2 Drivers for future proglacial channel avulsion


Glacifluvial sediment reworking of glacial deposits reduces landform connectivity and leads to a progressive stabilizing of
proglacial areas (Lane et al., 2016). Connectivity in turn is crucial for sediment storage or export (Bakker et al., 2018;
Geilhausen et al., 2012b). While proglacial lakes, like the 'Sandersee' or 'Pasterzensee' (Fig. 1), act as sediment traps (e.g.,
Bogen et al., 2014; Geilhausen et al., 2013; Krainer & Poscher 1992), the melt-out of (buried) dead ice landforms are still a
hidden effect on proglacial channel evolution, especially in recently deglaciated areas (e.g., Avian et al., 2018; Delaney et al.,
2018; Lane et al., 2016). In contrast to flood-driven river avulsion (e.g., Slingerland & Smith, 2004; Jones & Schumm, 1999;
Brizga & Finlayson, 1990), proglacial channel avulsion may be caused by the downwasting of dead ice landforms (e.g., Benn
& Evans, 2013; Lukas, 2011; Bennett and Glasser, 2009; Lukas et al., 2005; Richardson & Reynolds, 2000). Furthermore, the
melt-out of buried dead ice beyond the channel in the subsurface can result in channel bed settlement. However, this process
will not change the sediment composition of the erosion-resistant pavement layer of non-fluvial sediment. One prerequisite
for the development of such landforms is debris-covered glacier surface (Benn & Evans, 2013), whose progressive increase
can be observed worldwide (Mayr & Haag, 2019), in Europe (Lardeux et al., 2016), and thus also at the glacier Pasterze
(Fischer et al., 2018). Consequently, different dead ice landforms like hummocky moraines, ice-cored moraines, or kettles
could be detected at the foreland of glacier Pasterze (e.g., Avian et al., 2018; Geilhausen et al., 2012b; Krainer & Poscher,
1992). Investigating channel evolution in response to melting dead ice landforms is highly relevant to (i) describing future
proglacial channel development and (ii) quantifying proglacial sediment yields and sediment dynamics.

## 6 Summary and Conclusion


The distinction and transition from armor layers to erosion-resistant pavement layers by non-fluvial sediment is an important
definition and process in the proglacial channel evolution. Triggered by runoff variability due to global warming, the
establishment of non-erodible pavement layers is an essential post-glacial development process and has been widely neglected
up to now in defining proglacial channel evolution stages.
(1)   While recently deglaciated river sections are prone to glacifluvial headward erosion (against flow direction parallel to
glacier retreat) due to the fine sediment composition of the outwash plain, river sections with a greater distance to the
glacier terminus are characterized by sediment coarsening. This gradual process will limit further channel bed incision
by establishing an erosion-resistant pavement layer by non-fluvial deposit. Triggered by global warming, the short-term
increase and long-term decrease of the flow competence will develop pavement layers, which are in contrast to
infrequently mobile armoring layers. This development is considered as a final process in proglacial river evolution.
(2)   The calculation results of non-fluvial deposits forming pavement layers allow the extension of the proglacial sediment
cascade model by a new in-stream storage type within the fluvial system. This extension results in a refinement of the
existing fluvial part of the cascade approach: (i) braided channels in direct glacier proximity differ from (ii) the 'developed





channels' with increasing distance to the glacier terminus. This development leads to vertical landform decoupling
between the erosion-resistant proglacial channel bed and the unconsolidated diamictic sediment in the subsurface.
(3) In the long-term perspective, river avulsion driven by the melt-out of (buried) dead ice landforms will mainly contribute
to the stabilization in the catchment and reach scale. Investigating the channel evolution in response to melting dead ice
landforms is highly relevant for quantifying future post-glacial sediment dynamics. However, due to the characteristics
of glacial diamicton (poorly sorted sediment matrix ranging in size from sand up to boulders), proglacial channel
evolution will always lead to the final stage of pavement layer formation, as proven and described in this study.

**Data availability**

All the experimental data used in this study are available from the authors upon request.

**Author contribution**

MP and CH planned and designed the research. MP, PF, AN, GW, and BH performed the investigation, data curation, and
evaluation. The original draft preparation and visualization were done by MP and CH with equal contributions from all co-
authors. All authors were part of the review and editing of the manuscript.

**Competing interests**

The authors declare that they have no conflict of interest.

**Disclaimer**

Publisher's note: Copernicus Publications remains neutral with regard to jurisdictional claims in published maps and
institutional affiliations.

**Acknowledgment**

This paper was written as a contribution to the Christian Doppler Laboratory for Sediment Research and Management. In this
context, the financial support by the Christian Doppler Research Association, the Austrian Federal Ministry for Digital and
Economic Affairs and the National Foundation for Research, Technology and Development is gratefully acknowledged.
Moreover, the authors thank Rolf Rindler and Martin Fuhrmann for supportive fieldwork and Johann Aigner for discussions
on sediment transport dynamics.





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
