# Peer review of "Channel evolution processes in a diamictic glacier foreland. Implications on downstream sediment supply: case study Pasterze / Austria"

_Hydrology and Earth System Sciences, 2022_

## Referee Comment (RC1)

Review of Paster et al.: "Channel evolution processes in a diamictic glacier foreland. Implications on downstream sediment supply: case study Pasterze / Austria"

Manuscript ID: hess-2022-347

In their manuscript, Paster et al. investigate the channel evolution of a proglacial stream draining the Pasterze glacier in the Austrian Alps between 2015 and 2018. Using a combination of field surveying, remote sensing imagery and hydrodynamic modelling, the authors investigate the recent and future evolution of the sediment transport along the ~850 m long river reach. Relying on predicted runoff until 2050, the authors model the future transport capacity of the river and compare this to field and remote sensing derived measurements of grain size distributions. From their analysis, the authors conclude that the continuous erosion of finer sediments leaves very coarse grain sizes in the channel, armouring the bed and ultimately stabilizing the proglacial river system.

The manuscript is well written and addresses a topic that is potentially interesting for a broad range of readership. Before acceptable for publication in Hydrology and Earth System Sciences, however, the authors need to address a number of general and specific issues that I outlined in detail below.

General comments

A) Introduction: While the introduction is generally well written, it would certainly benefit from a stronger focus on the research gap that the authors want to address in their study. In my view, the aims should be better linked to the scientific context presented in the introduction. This is also true for the relevance and importance of the study. The authors are addressing the fundamentally important topic of how sediment dynamics might change in a changing climate. This should be stressed more explicitly, especially in a setting where the Margaritzenstausee reservoir is located only a kilometre downstream of the study site. Managing the sediment influx of reservoirs is a big issue that calls for studies that enhance our understanding of proglacial sediment dynamics.

B) Sediment sampling: Applying two different methods, line-by-number sampling in the accessible river sections and manually measuring grains in orthomosaics in the inaccessible canyon, respectively, makes comparability of the data generated an issue. It remains unclear why the authors did not construct partial grain size distributions from the same method in the entire study area. Given the inaccessibility of the canyon, visually measuring grains in orthomosaics would be suited for this and assure comparability. Another way forward would be to construct partial grain size distributions from both methods for some of the sampling points to assess the difference between methods and quantify a potential bias towards larger grain sizes introduced by measuring clasts in the orthomosaics. The authors should explicitly address the uncertainties introduced by the application of two different methods in their manuscript. The studies cited in L144-145 would suggest a shift towards larger grain sizes. Moreover, the description of the photogrammetrical sampling would benefit from a more detailed description. A final point to consider would be to make use of the automated extraction of grain size distributions from images (photo sieving) that also has been applied to entire reaches (Purinton and Bookhagen 2019). This would allow the authors to construct a more complete data set on grain size distributions by increasing their sampling size (only ten locations so far).

C) Evolution of the river reach: Despite the interesting data set the authors present and analyse in the present manuscript, some interpretations in the manuscript are not fully backed by the data and some discussion points do not cover all relevant aspects. Multiple studies (also

cited in the manuscript) show proglacial areas to be highly dynamic systems, where changes can happen within single events. Here the authors use an orthophoto from 2015 (from the federal government) and a UAV derived orthoimage from 2018 to investigate the channel dynamics. But this data set does not allow insight with higher temporal resolution, e.g. the interpretation that the bed is actually stabilizing. Here, the authors should make use of additional data sets that might allow a detailed quantification of the short-term dynamics. Furthermore, the interpretation that glacifluvial erosion in the channel leads to bed armouring and ultimately to a stabilized channel is based on the assumption that a) the channel does not migrate laterally or completely changes course and b) that sediment delivery to the channel reach does not change dramatically. Below I added comments in this respect to specific locations.

Specific comments

- L14: analyses instead of analysis
- L32: "steadily increasing spatial boundary"? Does this refer to the proglacial area that increases due to glacial recession? Consider rephrasing
- L36-39: consider splitting in two sentences
- L40: maybe: be described as a sediment cascade?
- L55-56: River bed incision into glacifluvial sediment and the formation of an armour layer is portrayed here as inevitable. While this might be true on the long run (when the catchment is devoid of transportable sediment), lateral migration, sediment delivery from valley flanks and a complete sift of the channel can happen in highly dynamic proglacial environments.
- L59: is able to transport sediments
- L65-66: For catchments with smaller glaciers, this peak-water effect has probably already been crossed, whereas for larger glaciers, this still lies in the future.
- L66-68: the second part of this sentence is not clear, please rephrase
- L69: repetition of L44-45
- L79-86: I think it would be important to mention that the reservoir Margaritzenstausee is located directly downstream. This increases the relevance of the study, as sediment management is an important topic for the reservoir.
- L91: Please explain the abbreviations here and elaborate how these values have been calculated. Was the length measured as Euclidean distance between start and end point of the segments, or along the channel? This also applies for the calculation of channel slope that can be derived from digital topographic data in multiple ways.
- L97-99: explain abbreviation "LbN" in the figure caption and provide details on the coordinate reference system used in the figure.
- L105-106: Glacifluvial processes are an important process for paraglacial adjustment, I am unsure why paraglacial reworking is contrasted here with glacifluvial processes?
- L124: Indicate which version of Agisoft Photoscan (Metashape since some years) was used for processing
- L125: add reference
- L134-135: Here it is unclear what the 478231187 points refers to? Usually, a DEM is a 2D raster with a certain pixel size. Please add details on the ground resolution of the DEM and Orthomosaic here
- L140-141: incomplete sentence
- L143: partial grain size distributions

- L151-153: Please, can the authors add more detail on the data set and method by Schöner et al. (2013)? As this is a crucial input for the study, the readers will want to understand how the Glacier Runoff Evolution Model (GREM) works. Also, please add more detail to the reference Schöner et al. (2013) as cited in the manuscript. Searching for this reference I can only find a presentation on the EURAS-CLIMPACT project that does not contain any detail on the GREM.
- L152: GREM?
- L153-154: It there a reason why the high-resolution digital elevation model derived from UAV imagery cannot be used for a roughness determination here?
- L167-169: Might this data be subject to underestimation/overestimation as the clasts are not lying flat on the ground with their b-axis visible?
- L193-195: this is a decrease by factor two, but not by two orders of magnitude
- L199: what are "big roughness elements"?
- L204-206: Delete "so-called" as knickpoint (or knickzone) is a standard geomorphic term. Maybe add a small explanation here: […] knickpoint, a pronounced convexity in the longitudinal channel profile, […]
- L220-221: But as Fig. 5 shows, the channel has moved considerably in the three years between 2015 and 2018. Except for a few meters, the entire channel shifted considerably, in some locations more than ~50m. I agree that this dynamics are to be expected, as the channel is actively incising. If the authors really want to show that channel migration is lower in 2018 than in 2015 (which again can be expected), they need to show this by data. The automated imagery might help to quantify channel mobility over time.
- L221-222: As this area is highly dynamic, I am not sure whether these changes can be attributed to upstream controls. The collapsing front of the debris-covered glacier changes takes away the lateral confinement in this area and the channel can turn to a steeper course and incise (see August 2016 and August 2017 in Fig. 5a).
- L222-224: Also here, I am not convinced that the data presented support this claim. The authors use the 2015 orthophoto and the 2018 UAV derived data here. From these two points in time, lateral changes in the channel can only be quantified for the entire three years long interval. The lateral confinement by "debris-covered dead ice landforms" towards the south is crucial in this setting. It can be anticipated that in a few years from now, the channel will not be active anymore, but will have shifted towards the centre of the valley. This can already be seen in satellite imagery from the summer of 2022 (see Figs. R1 and R2 below).
- Figure 5b: It is hard to tell the difference between the lines indicating the start and end of the study area and the beginning of the canyon. Maybe colorize? Also in the legend, label should be "start of canyon" or "beginning of canyon"
- L231-232: Again, this is a bold claim relying on only two points in time. In my view, this would require a thorough quantification of channel dynamics with high temporal resolution.
- L233-235: The knickpoint is located in a conspicuous position at the left lateral margin of the valley. From the picture in Fig. 6 one gets the impression that bedrock is exposed in this specific situation. This would strongly limit the mobility of the knickpoint and limit its potential for headward erosion. Can the authors give more detail on the specific setting of the knickpoint?
- L235: This is the first time since the abstract (L15) that river bathymetry is mentioned. Please elaborate in the introduction, methods, and results section how and why river bathymetry was measured and what this adds to the study.
- L237-239: While there might be a tendency of river channels to be more stable in greater distance from the glacier terminus, other factors, most importantly channel slope, are playing a crucial role as well.

- L241-242: If this knickpoint is produced by underlying bedrock, knickpoint migration will be very slow.
- L244: "non-fluvial sediment"? In extreme cases, steep rivers can transport large blocks... I guess the authors want to make the claim that these sediments are glacially deposited and remain in position, while the finer clasts are eroded and transported by the river?
- L247: before, the formation of the canyon has been described as glacifluvial, why are the authors using the term "post-glacial" here?
- L250-261: But this stabilization "from a hydraulic point of view" (L252) or the establishment of "an erosion-resistant pavement layer" will only happen under the assumption that the channel will not migrate laterally, or even shift to a new course. Baewert and Morche (2014) show that in a proglacial environment of the Gepatschferner the channel completely shifted to a new course following one extreme precipitation event. Proglacial areas are highly dynamic, and this is especially true for their upper margin where melting dead ice constantly reshapes the topography of the valley floor.
- L257-261, L263-265, Figure 6: It is not clear to me how the authors a) define and b) predict these "erosional breakpoints"? Are these "erosional breakpoints" not identical to the locations where partial grain size distributions were constructed from UAV derived imagery? If so, how can breakpoints (i.e. locations where something changes in my understanding) be defined based on six sample locations?
- L273-275: inevitable? I don't think this claim is justified, as I outlined before. Again, Baewert and Morche (2014) show an alteration from single thread to braided and back to single thread over a couple of years in a similar setting.
- L300-316: Also in this section the authors should attribute the various other possibilities of how the channel surveyed here might evolve in future.
- L318-337: Given all the concerns raised above, I would recommend the authors to formulate the conclusions much more cautiously here. While proglacial rivers might have a general tendency to stabilize due to bed armouring and the ultimate formation of a pavement layer, a lot of disturbances will distort this trajectory in a highly dynamic environment. Their survey of a single proglacial river section over the course of three years does not justify very general claims on the evolution of proglacial rivers.

References:

- Purinton, B. and Bookhagen, B.: Introducing *PebbleCounts*: a grain-sizing tool for photo surveys of dynamic gravel-bed rivers, Earth Surf. Dynam., 7, 859–877, https://doi.org/10.5194/esurf-7-859-2019, 2019.

[Figure]

Fig. R1 – Planet satellite image of the study area (2018-10-22). This is the situation as described in the manuscript. www.planet.com

[Figure]

Fig. R2 – Planet satellite image of the study area (2022-10-07). Note the formation of an incipient channel in towards the southwest of the old canyon. Future melting of dead ice will likely allow water flow in the center of the valley. Also note how the meander in the lower part of the channel changed its course. www.planet.com

---

## Referee Comment (RC2)

April 2023 review of Paster et al.: 'Channel evolution processes in a diamictic glacier foreland. Implications on downstream sediment supply: case study Pasterze / Austria'

In this manuscript, the authors aim to establish the dynamics and future trajectory of a glacier foreland. They place this case-study in the context of global warming and a key conceptual framework (the sediment cascade approach). Topographic data and surface grain size distributions were used to numerically model hydraulics and bedload transport. In conjunction with projections of future glacial runoff until 2050, these analyses were used to estimate current processes and predict the evolution of the foreland channel. The authors predict the erosion of finer sediments will lead to armouring and stabilisation of the channel, and propose improvements to the sediment cascade approach.

I will mention that this manuscript is not from my immediate field of expertise, however, I have provided comments as a fluvial geomorphologist on what may be required for this manuscript to be satisfactory for publication. At this point, I cannot recommend this for publication as it requires significant work in order to be suitable, and I recommend it be resubmitted in a more advanced form. This is a topic that is potentially interesting to the readership, and consequently I have provided general comments and recommendations that address the current limitations of the manuscript. I encourage the authors to carefully address the points below, and I am of course willing to re-evaluate this manuscript once this has been achieved.

General comments and recommendations

I have two general criticisms here, although they are related. First, there are insufficient data to draw strong process-based conclusions, let alone predictions about the future evolution of the system. This is made more challenging by this being a case study, which would require an especially high-quality dataset in order to contribute to the literature (and to a well-established conceptual framework). This can be broken down into three aspects:

- From my reading, UAV and sediment sampling were conducted in 2018, there was a comparison with a 2015 orthophoto, and over some time photos taken from an automatic camera. This provides a limited temporal comparison but also there is also little contextualisation of these data. More data are required on the history of this area to establish the oscillations (for example, seasonal and annual) at this site as well as its evolution over time. This is particularly important for the study as proglacial environments are highly dynamic over several timescales.
- The manuscript details that different surface grain size methods were used at different sites, but I could not see a comparison of these methods at the same site. This undermines the comparison. There needs to be a more convincing demonstration that differences in surface grain size distributions (Figure 3) are not simply due to differences in the sampling methodology. This may be more easily addressed compared to the above point.
- There was a reasonable quantification of error for the DEM preparation. However, there was not an adequate quantification or qualification or error and uncertainty in other

measurements, notably sediment sampling, hydraulic modeling, and bedload transport estimates. This makes it difficult to assess the results, for example, the predicted mobile D50 vs measured D50 present at the site.

Second, the link between the actual research conducted at the site and proposed improvement to the sediment cascade approach is tenuous. This site may offer some insights into such a conceptual model, although the current dataset and analyses do not currently allow for this due to the reasons outlined above. Lastly, at several points the language and expression need to be revised throughout for polish and clarity about the research findings. However, this is mostly editing, and can be resolved after the above points have been addressed.

Specific comments

L40 (approx) - this introduction paragraph should be divided up for readability.

L60 – comment about transportability of sediment is basically correct but lacks nuance surrounding partial mobility. Flow competence is important but the largest grain size fraction that is transported is transported most infrequently, so I would note the presence of partial transport and mention relevant literature (Wilcock & McArdell 1993 & 1997).

L64 – the word 'will' is used habitually throughout the manuscript when referring to projections of climate change and glacial discharge regimes. These are ultimately predictions and language should reflect this.

L70 – there is no clear research gap or problem that has been communicated. More generally, there needs to be a hypothesis or research question that is tested. Developing this will help in linking up analysis, discussion, and conclusions.

L76 – reference to a 'landform decoupling'. It is not clear what this is exactly, and similar process interpretations throughout need to be explained in specific terms.

L105 – These two processes of reworking are related, however.

L105-108 – Some of these statements about the dynamics of this site and potential for different processes are presented as rather factual, when they appear to be based on 1-2 studies. It may be useful to talk about these key studies and their methodologies so it's clear what has been demonstrated and how (e.g. Geilhausen et al., 2012b).

L139 – How was >150 stones decided? There are several rules-of-thumb across the literature, and this is not necessarily insufficient, however, some recent work has attempted to improve sampling and introduce a quantification of uncertainty. I will note one such effort by Eaton et al. 2019 that would help to demonstrate differences between GSDs more convincingly.

L147 – Usage of a 1D model should be justified given there is good drone data and there are likely important lateral processes occurring here which cannot be accounted for without a 2D approach.

L152 – there is no detail provided here regarding GERM, and this is needed for the study to be reproducible

L170 – There needs to be more detail here regarding both the orthophoto and the automatic camera. Especially with the limited temporal resolution of the dataset, the timing of these captures is critical.

L177 – narrowly graded, based on what criteria? There are indices to indicate the degree of gradation. I would be surprised if a proglacial stream was narrowly graded!

L180 – to me, describing these points as 'characteristic' would imply they are representative of whatever process are of interest. They seem to have been more arbitrarily selected

Figure 3: how has the 'potential future grain size distribution' been developed?

L196 - What is the justification for using D50 as the characteristic grain size? In gravel-bedded streams, a larger-than-average grain size percentile is usually more appropriate as it has greater influence over bedload transport (see Mackenzie et al. 2018).

L198-204: - It is difficult to assess the difference between these values without an estimate of uncertainty and error across the methods. Is a predicted mobile D50 diameter of 60 mm bigger than an observed surface D50 diameter of 50 mm in a way that is statistically significant? Sediment transport equations are not known for their high accuracy. Also, what does 'no big roughness elements' mean? Moreover, the assumption that the surface grain size distribution is representative of the sediment load needs to be more carefully addressed.

L219 – what does 'pronounced river structure' mean?

L237-239 – What is the evidence for this? How can one be sure this will occur at this site? The authors should be careful in the discussion to be clear what evidence there is for the specific area of study, compared to studies of other areas. There also needs to be discussion of what is meant by 'channel stability', as this varies widely.

Figure 6: - How have these 'erosion breakpoints' been defined? They appear remarkably spatially periodic; do they have physical meaning or are they just points that have selected for analysis?

L296-299 – there is some attempt here to discuss the limitations of the methodology, but this deserves a more comprehensive effort to demonstrate that these limitations do not undermine the findings. Relatedly, why is it acceptable that the relatively finer fractions were not accounted for? There is a large literature on the importance of fine sediment for decreasing the entrainment threshold of larger grains. What are the limitations of only sampling surface grains, as opposed to the bulk?

Section 5.2 and Conclusion: I find these sections unconvincing because they are largely unrelated to the data that has been presented. They appear to summarise general findings from the literature rather than detail the empirical and theoretical contributions of the study. The final clause 'as proven and described in this study' is inappropriate. This is, however, mostly an exercise in adjusting the conclusions to better reflect the work that's been done. I encourage them to think carefully through this process!

References

Eaton, Brett C., R. Dan Moore, and Lucy G. MacKenzie. "Percentile-based grain size distribution analysis tools (GSDtools)–estimating confidence limits and hypothesis tests for comparing two samples." Earth Surface Dynamics 7.3 (2019): 789-806.

MacKenzie, L.G., Eaton, B.C. and Church, M., 2018. Breaking from the average: Why large grains matter in gravel-bed streams. Earth Surface Processes and Landforms, 43(15), pp.3190-3196.

Wilcock, P.R. and McArdell, B.W., 1993. Surface-based fractional transport rates: Mobilization thresholds and partial transport of a sand-gravel sediment. Water Resources Research, 29(4), pp.1297-1312.

Wilcock, P.R. and McArdell, B.W., 1997. Partial transport of a sand/gravel sediment. Water Resources Research, 33(1), pp.235-245.

---

## Author Comment (AC1)

**Reply on anonymous Reviewer #1**

Review of Paster et al.: *"Channel evolution processes in a diamictic glacier foreland. Implications on downstream sediment supply: case study Pasterze / Austria"*
Manuscript ID: hess-2022-347

In their manuscript, Paster et al. investigate the channel evolution of a proglacial stream draining the Pasterze glacier in the Austrian Alps between 2015 and 2018. Using a combination of field surveying, remote sensing imagery and hydrodynamic modelling, the authors investigate the recent and future evolution of the sediment transport along the ~850 m long river reach. Relying on predicted runoff until 2050, the authors model the future transport capacity of the river and compare this to field and remote sensing derived measurements of grain size distributions. From their analysis, the authors conclude that the continuous erosion of finer sediments leaves very coarse grain sizes in the channel, armouring the bed and ultimately stabilizing the proglacial river system.

The manuscript is well written and addresses a topic that is potentially interesting for a broad range of readership. Before acceptable for publication in Hydrology and Earth System Sciences, however, the authors need to address a number of general and specific issues that I outlined in detail below.

> **Reply:** *Many thanks to the Reviewer for critically reviewing the submitted manuscript. We agree that the initial version of the manuscript (MS) needs a revision regarding the points the reviewer made. A detailed discussion of the comments can be found in the following section. We believe that we have considered all comments appropriately.*

**1. General comments**

**A)** Introduction: While the introduction is generally well written, it would certainly benefit from a stronger focus on the research gap that the authors want to address in their study. In my view, the aims should be better linked to the scientific context presented in the introduction. This is also true for the relevance and importance of the study. The authors are addressing the fundamentally important topic of how sediment dynamics might change in a changing climate. This should be stressed more explicitly, especially in a setting where the Margaritzenstausee reservoir is located only a kilometer downstream of the study site. Managing the sediment influx of reservoirs is a big issue that calls for studies that enhance our understanding of proglacial sediment dynamics.

> **Reply:** *(1) Thank you for this valuable comment. As stated in the initial MS, the gradual channel evolution (glacifluvial sediment reworking; LN55) and landform decoupling (LN42) are decisive parameters regarding proglacial landscape stabilization.*
> *(2) The predicted (i) establishment of an erosion-resistant pavement layer (ii) will force landform decoupling in the vertical direction with (iii) implications on the fluvial system of the proglacial sediment cascade model (key results of the submitted MS). This gradual development of proglacial rivers by glacifluvial sediment reworking has been neglected up to now in different approaches but is an urgent issue when describing (stabilization) processes of proglacial areas.*

*(3) We agree with the reviewer that the knowledge of the Margaritze reservoir downstream of the study site is informative regarding high-alpine sediment management. Thus, we added more information in the revised MS (see other replies).*

**B)** Sediment sampling: Applying two different methods, line-by-number sampling in the accessible river sections and manually measuring grains in orthomosaics in the inaccessible canyon, respectively, makes comparability of the data generated an issue. It remains unclear why the authors did not construct partial grain size distributions from the same method in the entire study area. Given the inaccessibility of the canyon, visually measuring grains in orthomosaics would be suited for this and assure comparability. Another way forward would be to construct partial grain size distributions from both methods for some of the sampling points to assess the difference between methods and quantify a potential bias towards larger grain sizes introduced by measuring clasts in the orthomosaics. The authors should explicitly address the uncertainties introduced by the application of two different methods in their manuscript. The studies cited in L144-145 would suggest a shift towards larger grain sizes. Moreover, the description of the photogrammetrical sampling would benefit from a more detailed description. A final point to consider would be to make use of the automated extraction of grain size distributions from images (photo sieving) that also has been applied to entire reaches (Purinton and Bookhagen 2019). This would allow the authors to construct a more complete data set on grain size distributions by increasing their sampling size (only ten locations so far).

*Reply: (1) First, the accuracy of the applied methods – partial grain size distribution by the line-by-number approach (Fehr, 1987) – is sufficient for the study hypothesis as we are interested in big (non-fluvial) grain sizes and their effect on proglacial channel evolution.*

*(2) Generally, sediment analysis is challenging in high-alpine, often inaccessible areas (initial MS – LN293). As the initial MS (LN137) also mentioned, the line-by-number method is a common field measurement method for gravel-to-cobble-bed mountain rivers (e.g., Fehr, 1987; Lang et al., 2021).*

*(3) However, we agree with the reviewer that photo-sieving can be more accurate when you have a high-resolution orthomosaic available. For photo-sieving, the ground sample distance (GSD) is the relevant parameter (stated in the initial MS – LN145), also mentioned by the proposed reference (Purinton and Bookhagen, 2019). Despite the high-resolution orthomosaic processed in this study (0.7 mm px$^{-1}$; see later reply), photo-sieving in the headwater was impossible due to the fine sediment composition. Thus, field measurements were invertible in this section. Nevertheless, line sampling was also applied in the digital approach to the inaccessible canyon to ensure comparability. As it is stated by Lang et al. (2021), "digital line sampling is the one-to-one counterpart of the current state-of-the-art field method [line-by-number analysis]" for mountain rivers.*

**C)** Evolution of the river reach: Despite the interesting data set the authors present and analyse in the present manuscript, some interpretations in the manuscript are not fully backed by the data and some discussion points do not cover all relevant aspects. Multiple studies (also cited in the manuscript) show proglacial areas to be highly dynamic systems, where changes can happen within single events. Here the authors use an orthophoto from 2015 (from the federal government) and a UAV derived orthoimage from 2018 to investigate the channel dynamics. But this data set does not allow insight with higher temporal resolution, e.g. the interpretation

that the bed is actually stabilizing. Here, the authors should make use of additional data sets that might allow a detailed quantification of the short-term dynamics. Furthermore, the interpretation that glacifluvial erosion in the channel leads to bed armouring and ultimately to a stabilized channel is based on the assumption that a) the channel does not migrate laterally or completely changes course and b) that sediment delivery to the channel reach does not change dramatically. Below I added comments in this respect to specific locations.

> *Reply: (1) We agree that proglacial areas are very dynamic systems (e.g., LN43, LN47 in the initial MS), also valid for the Pasterze landsystem (LN107 in the initial MS). But the special situation of the glacier forefield (channel confinement by the debris-covered glacier) hence lateral migration and force channel bed incision (besides other parameters mentioned in the initial MS – LN51).*
>
> *(2) We also agree with the reviewer that single events (like high-magnitude/low-frequency events) are drivers for changes in the proglacial channel pattern (e.g., LN49 or LN291 in the initial MS). Thus, armor layer formation and the establishment of an erosion-resistant pavement layer in the long term do not stabilize the entire channel.*
>
> *(3) The intention of the comparison between 2015 and 2018 should show that lateral migration is limited due to the confinement by the debris-covered glacier. Thus, the study results do not allow a conclusion regarding the entire proglacial channel stabilization but give a tendency of an erosion-resistant pavement layer establishment and thus on channel bed stabilization. During revision, the focus was made on more precise wording describing this gradual development process of the proglacial river bed.*
>
> *(4) Due to the characteristic sediment composition of a glacial outwash plain (diamictic till), the gradual development by glacifluvial sediment reworking to an erosion-resistant pavement layer is given, even when lateral migration and continuous sediment supply will occur in the future.*

**2. Specific comments**

**L14:** analyses instead of analysis
> *Reply: Thank you, we corrected this sentence.*

**L32:** "steadily increasing spatial boundary"? Does this refer to the proglacial area that increases due to glacial recession? Consider rephrasing
> *Reply: Correct, the increase of the proglacial area due to glacier retreat is meant by this sentence. We rephrased this sentence during the revision.*

**L36-39:** consider splitting in two sentences
> *Reply: Thank you, we rephrased this part as suggested.*

**L40:** maybe: be described as a sediment cascade?
> *Reply: Thank you, we rephrased this sentence as suggested.*

**L55-56:** River bed incision into glacifluvial sediment and the formation of an armour layer is portrayed here as inevitable. While this might be true on the long run (when the catchment is devoid of transportable sediment), lateral migration, sediment delivery from valley flanks and a complete sift of the channel can happen in highly dynamic proglacial environments.

*Reply: Thank you for this valuable comment. We agree with the reviewer that proglacial environments are highly dynamic areas (see also previous reply) due to processes mentioned by the reviewer (e.g., lateral migration, channel avulsion). That river bed incision or a stabilized river bed by non-fluvial sediment (study results) does not prevent lateral migration is already mentioned in the discussion of the initial MS (LN290). We considered that in more detail and mention this fact also in the introduction of the revised MS.*

*(2) An established erosion-resistant pavement layer will remain intact, despite sediment supply by, e.g., lateral migration (like river embankment failure) or sediment input by valley flanks. Thereby, landform coupling is decisive (LN41 in the initial MS). In the investigated study area, the lateral valley flanks (hillslope-channel connection) are decoupled from the fluvial system. Thus, only glacial deposited material is transported glacifluvial downstream (LN103 in the initial MS; Geilhausen et al., 2012). We considered all this information in more detail in the revised MS.*

**L59:** is able to transport sediments

*Reply: Thank you, we revised it as suggested.*

**L65-66:** For catchments with smaller glaciers, this peak-water effect has probably already been crossed, whereas for larger glaciers, this still lies in the future.

*Reply: (1) Thank you for this comment. It is correct that the moment of peak water in many glacier-fed rivers has already passed, especially in mountain regions with smaller glaciers (Hock et al., 2019). Huss and Hock (2018) suggest that peak water has been reached for up to 67 % of central European glaciers (including the European Alps). As it is mentioned in the initial MS (LN65), the peak water is suggested to be reached for the European Glaciers latest around the middle of the century (Huss and Hock, 2018). We added additional information during the revision.*

**L66-68:** the second part of this sentence is not clear, please rephrase

*Reply: Thank you, we rephrased this sentence.*

**L69:** repetition of L44-45

*Reply: We deleted the last part of this sentence.*

**L79-86:** I think it would be important to mention that the reservoir Margaritzenstausee is located directly downstream. This increases the relevance of the study, as sediment management is an important topic for the reservoir.

*Reply: We agree with the reviewer that knowledge about the reservoir Margaritze is important (see previous reply). Thus, we include this information during revision.*

**L91:** Please explain the abbreviations here and elaborate how these values have been calculated. Was the length measured as Euclidean distance between start and end point of the segments, or along the channel? This also applies for the calculation of channel slope that can be derived from digital topographic data in multiple ways.

*Reply: (1) The distance between the respective cross-sections (CS) was measured along the thalweg (the lowest point in each CS).*

*(2) The gradient of the 'trend line' (based on the lowest point in each CS) was used for the gradient calculation.*

**L97-99:** explain abbreviation "LbN" in the figure caption and provide details on the coordinate reference system used in the figure.

*Reply: We implemented the missing information in Fig. 1 during the revision.*

**L105-106:** Glacifluvial processes are an important process for paraglacial adjustment, I am unsure why paraglacial reworking is contrasted here with glacifluvial processes?

*Reply: Thank you for this comment. Glacifluvial sediment reworking and paraglacial reworking are indeed not contrasting processes. Thus, we rephrased this section.*

**L124:** Indicate which version of Agisoft Photoscan (Metashape since some years) was used for processing.

*Reply: We used Agisoft Photoscan version 1.2.6 for processing the DEM and implemented this information in the revised MS.*

**L125:** add reference

*Reply: We added the reference (Westoby et al., 2012) here.*

**L134-135:** Here it is unclear what the 478231187 points refers to? Usually, a DEM is a 2D raster with a certain pixel size. Please add details on the ground resolution of the DEM and Orthomosaic here

*Reply: (1) The point number is a meta information of Agisoft. We agree with the reviewer that this is redundant information, so we deleted it in the revised MS.*
*(2) The pixel size (ground sample distance GSD) of the DEM has already been specified in the initial MS (LN135). In the initial MS, we used the GSD for describing the ground resolution, which is now corrected in the revised MS.*
*(3) We added additional information about the ground resolution of the DEM (1.59 cm px$^{-1}$) and the orthomosaic (0.7 mm px$^{-1}$).*

**L140-141:** incomplete sentence

*Reply: Thank you, we improved this sentence during the revision.*

**L143:** partial grain size distributions

*Reply: Thank you, we revised it as suggested.*

**L151-153:** Please, can the authors add more detail on the data set and method by Schöner et al. (2013)? As this is a crucial input for the study, the readers will want to understand how the Glacier Runoff Evolution Model (GREM) works. Also, please add more detail to the reference Schöner et al. (2013) as cited in the manuscript. Searching for this reference I can only find a presentation on the EURAS-CLIMPACT project that does not contain any detail on the GREM.

*Reply: We added more information about the GERM in the revised MS as colleagues (GeoSphere Austria, formerly known as Central Institution of Meteorology and Geodynamics) were involved in processing GERM for the Pasterze Glacier and are co-authors of this study.*

**L152:** GREM?

*Reply: This hydrological model is called 'Glacier Evolution Runoff Model (GERM)', meaning the acronym in the initial MS was correctly spelled. Accordingly, we corrected the name of this hydrological model in the revised MS.*

**L153-154:** Is there a reason why the high-resolution digital elevation model derived from UAV imagery cannot be used for a roughness determination here?

*Reply: (1) As it is well summarized by Pearson et al. (2017), different approaches exist for roughness determination by a high-resolution DEM. The wide spreading results show the difficulty of applying this approach. Furthermore, almost all approaches deal with grain sizes clearly smaller than we investigated.*

*(2) For 1D hydrodynamic modeling, however, the manning value (reciprocal value of the Strickler value) is needed. The well-known formula according to Strickler (1923) – determination of $k_{st}$ by $d_{90}$ ($k_{st}= 26/d_{90}^{1/6}$) – is only valid for rivers (i) with ideal smooth river bed and (ii) with gravel and sand as bed material. However, this study mainly investigates a torrent with all grain sizes up to macro-roughness elements.*

*(3) Thus, a representative value was set based on sensitivity analysis and literature. According to Naudascher (1992), a $k_{st}= 19\text{-}22\ m^{1/3}s^{-1}$ should be used for torrents with macro-roughness elements and high active bedload transport. As the canyon of the investigated study reach matches with this torrent description, $kst=20\ m^{1/3}s^{-1}$ is appropriate for the canyon. For the headwater and delta, $k_{st}=28\ m^{1/3}s^{-1}$ is used as suggested in the literature for reaches without macro-roughness elements but with high active bedload transport.*

**L167-169:** Might this data be subject to underestimation/overestimation as the clasts are not lying flat on the ground with their b-axis visible?

*Reply: The big clasts in proglacial rivers are usually outwashed and exposed and thus clearly delineated. The missing knowledge of the third stone axis on sediment analysis based on images also applies to all other digital sediment analysis approaches like photo-sieving.*

**L193-195:** this is a decrease by factor two, but not by two orders of magnitude.

*Reply: Thank you for this valuable comment, we corrected it in the revised MS.*

**L199:** what are "big roughness elements"?

*Reply: (1) Big roughness elements like boulders or step-pool sequences create increased flow resistance in steep mountain streams and are important controls of bedload transport. (2) In literature (e.g., Nitsche et al., 2011), the more common term "macro-roughness elements" is used. Therefore, we adopted this more appropriate term in the revised MS.*

**L204-206:** Delete "so-called" as knickpoint (or knickzone) is a standard geomorphic term. Maybe add a small explanation here: […] knickpoint, a pronounced convexity in the longitudinal channel profile, […]

*Reply: Thank you for this comment, we revised it as suggested.*

**L220-221:** But as Fig. 5 shows, the channel has moved considerably in the three years between 2015 and 2018. Except for a few meters, the entire channel shifted considerably, in some locations more than ~50m. I agree that this dynamics are to be expected, as the channel is actively

incising. If the authors really want to show that channel migration is lower in 2018 than in 2015 (which again can be expected), they need to show this by data. The automated imagery might help to quantify channel mobility over time.

> *Reply: With this comparison, we want to show that lateral confinement has existed since the channel development's beginning. Pictures from the automated camera show a very stable channel since 2015 – explained by the very slow melting rate of the debris-covered glaciers – and a very dynamic delta area (as shown and described in Fig. 5). In the revised MS, we are more precise in wording.*

**L221-222:** As this area is highly dynamic, I am not sure whether these changes can be attributed to upstream controls. The collapsing front of the debris-covered glacier changes takes away the lateral confinement in this area and the channel can turn to a steeper course and incise (see August 2016 and August 2017 in Fig. 5a).

> *Reply: Disappearing lateral confinement leads to channel widening with increasing wetted width, lower water depth, and reduced shear stress.*

**L222-224:** Also here, I am not convinced that the data presented support this claim. The authors use the 2015 orthophoto and the 2018 UAV derived data here. From these two points in time, lateral changes in the channel can only be quantified for the entire three years long interval. The lateral confinement by "debris-covered dead ice landforms" towards the south is crucial in this setting. It can be anticipated that in a few years from now, the channel will not be active anymore, but will have shifted towards the centre of the valley. This can already be seen in satellite imagery from the summer of 2022 (see Figs. R1 and R2 below).

> *Reply: (1) We agree with the reviewer that dead ice melting will lead to lateral channel migration and channel avulsion (as already discussed in the initial MS in LN303-309).*
> *(2) As the channel is confined by debris-covered dead ice, the lateral changes strongly depend on dead ice melting. Due to this 'special environment', dead ice melting is more relevant as bank erosion for lateral migration and is now appropriately mentioned in the revised MS. Furthermore, collapsing dead-ice landforms may enable completely new channels.*
> *(3) This expected future development is more appropriately addressed in the revised MS.*

**Figure 5b:** It is hard to tell the difference between the lines indicating the start and end of the study area and the beginning of the canyon. Maybe colorize? Also in the legend, label should be "start of canyon" or "beginning of canyon"

> *Reply: (1) As figures should be readable in black/white, we revised this figure but used a black (and white, respectively) dotted line for the start of the canyon instead of colors.*
> *(2) We adopted the suggestions for labeling.*

L231-232: Again, this is a bold claim relying on only two points in time. In my view, this would require a thorough quantification of channel dynamics with high temporal resolution.

> *Reply: As mentioned in several replies, we used more precise wording during the revision - stabilization in terms of "channel bed stabilization"!*

**L233-235:** The knickpoint is located in a conspicuous position at the left lateral margin of the valley. From the picture in Fig. 6 one gets the impression that bedrock is exposed in this specific

situation. This would strongly limit the mobility of the knickpoint and limit its potential for headward erosion. Can the authors give more detail on the specific setting of the knickpoint?

*Reply: Thank you for this very valuable comment. Further analysis will be done on the knick-point condition and its future effect on the (upstream) channel development.*

L235: This is the first time since the abstract (L15) that river bathymetry is mentioned. Please elaborate in the introduction, methods, and results section how and why river bathymetry was measured and what this adds to the study.

*Reply: Bathymetry wasn't measured, as it was impossible due to inaccessibility. We revised the wording of this paragraph and the Abstract during the revision.*

L237-239: While there might be a tendency of river channels to be more stable in greater distance from the glacier terminus, other factors, most importantly channel slope, are playing a crucial role as well.

*Reply: We agree with the reviewer that (i) channel slope, (ii) sediment composition, and (iii) runoff variability are decisive for channel pattern changes and river stabilization (already mentioned in the initial MS - LN52). However, landform decoupling by different reworking processes with a greater distance to the glacier terminus is also a decisive parameter (e.g., Bakker et al., 2018; Geilhausen et al., 2012; Fryris et al., 2007).*

**L241-242:** If this knickpoint is produced by underlying bedrock, knickpoint migration will be very slow.

*Reply: Thank you for this comment, we agree with the reviewer and we are now more precise in the revised MS (see previous reply).*

**L244:** "non-fluvial sediment"? In extreme cases, steep rivers can transport large blocks… I guess the authors want to make thr7e claim that these sediments are glacially deposited and remain in position, while the finer clasts are eroded and transported by the river?

*Reply: Correct, non-fluvial sediments are those glacially deposited and never transported by hydraulic forces. These non-fluvial sediments mainly form the erosion-resistant pavement layer.*

**L247:** before, the formation of the canyon has been described as glacifluvial, why are the authors using the term "post-glacial" here?

*Reply: Thank you for this comment. During the revision, we focused on precise and consistent wording throughout the MS.*

**L250-261:** But this stabilization "from a hydraulic point of view" (L252) or the establishment of "an erosion-resistant pavement layer" will only happen under the assumption that the channel will not migrate laterally, or even shift to a new course. Baewert and Morche (2014) show that in a proglacial environment of the Gepatschferner the channel completely shifted to a new course following one extreme precipitation event. Proglacial areas are highly dynamic, and this is especially true for their upper margin where melting dead ice constantly reshapes the topography of the valley floor.

*Reply: (1) With the term stabilization, the channel bed is meant and not the cross-sectional shape. We had the focus on this precise wording during the revision (see previous reply).*

*(2) But we agree with the reviewer that proglacial areas are highly dynamic – as stated in the initial MS (e.g., LN43, LN47) – and our study results cannot predict the stabilization of entire proglacial channels.*

*(3) If lateral channel migration will occur, a lot of sediment is available for fluvial transportation (mentioned in LN290). But this fluvial sediment transport leads again to the armor-layer formation and later to the erosion-resistant pavement layer (see previous reply).*

**L257-261, L263-265, Figure 6:** It is not clear to me how the authors a) define and b) predict these "erosional breakpoints"? Are these "erosional breakpoints" not identical to the locations where partial grain size distributions were constructed from UAV derived imagery? If so, how can breakpoints (i.e. locations where something changes in my understanding) be defined based on six sample locations?

*Reply: (1) Correct, the defined 'erosion breakpoints' are identical to the locations of the partial grain size distributions in the canyon.*

*(2) The term 'breakpoint' is, per definition, a location where a continuous process is interrupted. Thus, we implemented the term 'erosion breakpoints' in this study, as the continuous river bed incision is stopped by the tendency of an establishment of an erosion-resistant pavement layer, examined exemplarily at those six characteristic points. There, the tendency of establishing an erosion-resistant pavement layer is given based on our study results.*

**L273-275:** inevitable? I don't think this claim is justified, as I outlined before. Again, Baewert and Morche (2014) show an alteration from single thread to braided and back to single thread over a couple of years in a similar setting.

*Reply: (1) We agree with the reviewer that the study results cannot conclude a strict delimitation between braided and single-thread river sections. Single events or the melt-out of dead-ice landforms can change the system again between supply- and transport-limited.*

*(2) However, after a possible channel avulsion in the future, the glacifluvial processes result again in the establishment of an armor layer leading in the long-term to an erosion-resistant pavement layer again.*

*(3) As mentioned in several replies before, we were more precise in the wording (the tendency of stabilized river bed) during the revision.*

**L300-316:** Also, in this section the authors should attribute the various other possibilities of how the channel surveyed here might evolve in future.

*Reply: Thank you for this comment. As stated in the initial MS (LN304 onwards), the melt-out of the buried dead-ice landforms is able for proglacial channel avulsion (see also previous reply). We added more information about possible future development on the studied channel at the Pasterze landsystem.*

**L318-337:** Given all the concerns raised above, I would recommend the authors to formulate the conclusions much more cautiously here. While proglacial rivers might have a general tendency to stabilize due to bed armouring and the ultimate formation of a pavement layer, a lot of disturbances will distort this trajectory in a highly dynamic environment. Their survey of a single proglacial river section over the course of three years does not justify very general claims on the evolution of proglacial rivers.

*Reply: (1) Thank you for this valuable comment. We agree with the reviewer that proglacial areas are very dynamic systems (e.g., LN43, LN47 in the initial MS; see previous reply).*
*(2) However, based on predicted discharge data by 2050, the study results show that non-fluvial sediment is already present at the bed of the channel in the canyon. These results claim the tendency of channel bed stabilization. Thus, the results justify the refinement and extension of the fluvial system of a proglacial sediment cascade.*

**References:**

Purinton, B. and Bookhagen, B.: Introducing *PebbleCounts*: a grain-sizing tool for photo surveys of dynamic gravel-bed rivers, Earth Surf. Dynam., 7, 859–877, https://doi.org/10.5194/esurf-7-859-2019, 2019.

*Reply: Thank you for this reference. As it is already stated in the initial MS (LN145), the lower truncation for adequate b-axis length measurements in a digital approach is strongly dependent on the ground sample distance (see previous reply): ≥20 px (Purinton and Bookhagen, 2019); ≥10-15 (Detert et al., 2018) or ≥4 px (Lang et al., 2021).*

**Fig. R1** – Planet satellite image of the study area (2018-10-22). This is the situation as described in the manuscript. www.planet.com

Fig. R2 – Planet satellite image of the study area (2022-10-07). Note the formation of an incipient channel in towards the southwest of the old canyon. Future melting of dead ice will likely allow water flow in the center of the valley. Also note how the meander in the lower part of the channel changed its course. www.planet.com

*Reply: (1) We agree with the reviewer that melting dead ice bodies are able to open up new channels. This potential process in the future was already mentioned in the revised MS (LN307-309). In the initial manuscript's Summary and Conclusion (LN333-335), we also mentioned the great relevance of melting dead ice landforms regarding proglacial channel evolution and stabilization of entire proglacial areas.*
*(2) We added more information about the (possible) future development of the proglacial area from the fluvial perspective during this revision (see previous reply).*

---

## Author Comment (AC2)

**Reply to anonymous Reviewer #2**

April 2023 review of Paster et al.: *'Channel evolution processes in a diamictic glacier foreland. Implications on downstream sediment supply: case study Pasterze / Austria'*

In this manuscript, the authors aim to establish the dynamics and future trajectory of a glacier foreland. They place this case-study in the context of global warming and a key conceptual framework (the sediment cascade approach). Topographic data and surface grain size distributions were used to numerically model hydraulics and bedload transport. In conjunction with projections of future glacial runoff until 2050, these analyses were used to estimate current processes and predict the evolution of the foreland channel. The authors predict the erosion of finer sediments will lead to armouring and stabilisation of the channel, and propose improvements to the sediment cascade approach.

I will mention that this manuscript is not from my immediate field of expertise, however, I have provided comments as a fluvial geomorphologist on what may be required for this manuscript to be satisfactory for publication. At this point, I cannot recommend this for publication as it requires significant work in order to be suitable, and I recommend it be resubmitted in a more advanced form. This is a topic that is potentially interesting to the readership, and consequently I have provided general comments and recommendations that address the current limitations of the manuscript. I encourage the authors to carefully address the points below, and I am of course willing to re-evaluate this manuscript once this has been achieved.

> **Reply:** *We thank the Reviewer for the critical and detailed review of the submitted manuscript (MS). We agree that the initial version of the MS needs a revision regarding the reviewer's points. A detailed discussion of the comments can be found in the following section. We are convinced that we have considered all comments appropriately.*

**1. General comments and recommendations**

I have two general criticisms here, although they are related. First, there are insufficient data to draw strong process-based conclusions, let alone predictions about the future evolution of the system. This is made more challenging by this being a case study, which would require an especially high-quality dataset in order to contribute to the literature (and to a well-established conceptual framework). This can be broken down into three aspects:

**A)** From my reading, UAV and sediment sampling were conducted in 2018, there was a comparison with a 2015 orthophoto, and over some time photos taken from an automatic camera. This provides a limited temporal comparison but also there is also little contextualization of

these data. More data are required on the history of this area to establish the oscillations (for example, seasonal and annual) at this site as well as its evolution over time. This is particularly important for the study as proglacial environments are highly dynamic over several timescales.

> *Reply: Thank you for this comment. We agree with the reviewer that proglacial areas are highly dynamic environments (initial MS – e.g., LN42, 47, 107, 220). However, special focus was given to the evidence of stabilization processes. Studies have shown stabilized channel sections with increasing distance to the glacier terminus (e.g., Gurnell et al., 1999).*
>
> *(1) Based on forecasted hydrological model data (glacier evolution runoff model – GERM), the study results give a prediction/tendency of the future channel bed stabilization in a well-known diamictic environment. Channel stability is strongly coupled to landform (dis-)connectivity (Fryris et al., 2007; initial MS – LN41). Elements that force vertical decoupling are non-fluvial sediment (initial MS – LN56), forming first an infrequent mobile armoring layer (Bunter & Abt, 2001), which transforms in longer term to a (more) erosion-resistant pavement layer. The hydraulic modeling results underline this gradual development, although uncertainties are given (initial MS – e.g., LN143, LN297; and later replies).*
>
> *(2) The comparison with 2015 (origin of the channel, initial MS – LN172) was done to get an overview of the study area since the channel formation. However, all study results are based on hydrological modeling (GERM). Care was taken in the review to clarify this difference to avoid misunderstanding (see later replies).*

**B)** The manuscript details that different surface grain size methods were used at different sites, but I could not see a comparison of these methods at the same site. This undermines the comparison. There needs to be a more convincing demonstration that differences in surface grain size distributions (Figure 3) are not simply due to differences in the sampling methodology. This may be more easily addressed compared to the above point.

> *Reply: Thank you for this important comment. We used the line-by-number method (Fehr, 1987) in two different ways: (i) on-site and (ii) digital in the processed high-resolution orthomosaic based on the UAV images. As stated by Lang et al. (2021), "the digital line sampling is the one-to-one counterpart of the current state-of-the-art field method [line-by-number analysis]".*
>
> *(1) A methodological comparison of both approaches is not possible due to (i) the very fine sediment composition in the headwater and (ii) the inaccessibility of the canyon.*
>
> *(2) We supplemented the already addressed study uncertainties (initial MS – LN296) with those mentioned by the reviewer (see other replies, respectively).*

**C)** There was a reasonable quantification of error for the DEM preparation. However, there was not an adequate quantification or qualification or error and uncertainty in other measurements, notably sediment sampling, hydraulic modeling, and bedload transport estimates. This makes it difficult to assess the results, for example, the predicted mobile D50 vs measured D50 present at the site. Second, the link between the actual research conducted at the site and proposed improvement to the sediment cascade approach is tenuous. This site may offer some insights into such a conceptual model, although the current dataset and analyses do not currently allow for this due to the reasons outlined above. Lastly, at several points the language and expression need to be revised throughout for polish and clarity about the research findings. However, this is mostly editing and can be resolved after the above points have been addressed.

*Reply: Thank you for this valuable comment; knowledge about uncertainties is essential (see previous reply).*

*(1) The river bed coarsening by glacifluvial erosion (pavement layer formation) for the presented case study is a continuous process over the upcoming years due to steadily increased bed shear stress strongly coupled to the predicted discharge (according to GERM until the year 2030; initial MS – LN192). This continuous process of river bed coarsening gradually increases the river bed's characteristic grain sizes (e.g., $d_{50}$). In our study results, the actual characteristic grain size $d_{50.m}$ (field measurement) in 2018 is already bigger than the flow competence (hydraulic model results) according to the highest predicted discharge in 2030 ($d_{50.c}$). This difference will increase in the future due to the gradual coarsening of the substrate of the channel bed. After the year 2030, the flow competence is expected to decrease around the level of 2018 (initial MS – LN193), which in turn limit the initiation of motion of bedload. As stated in a later reply, the tendency of pavement layer formation is still given considering uncertainties in sediment sampling.*

*(2) As described above, the (predicted) gradual (year-by-year) coarsening of the river bed underlines the study's results, which are supported from a hydraulic point of view. As this gradual development of a pavement layer is valid for areas of diamictic glacial till, the new "in-stream" storage type of non-fluvial sediment and, thus, the extension and refinement are valid for all other catchments with the same characteristics. The pavement layer is a relevant landform decoupling process (acting as buffers in the vertical direction – initial MS LN42, 284; Brierley et al., 2006), which is specific and poorly investigated in (diamictic) glacier forelands.*

*(3) We focused on precise formulation and consistent wording during the revision.*

**2. Specific comments**

**L40 (approx) -** this introduction paragraph should be divided up for readability.

*Reply: We revised it as suggested.*

**L60 –** comment about transportability of sediment is basically correct but lacks nuance surrounding partial mobility. Flow competence is important but the largest grain size fraction that is transported is transported most infrequently, so I would note the presence of partial transport and mention relevant literature (Wilcock & McArdell 1993 & 1997).

*Reply: Thank you for this additional information. We added it appropriately.*

**L64 –** the word 'will' is used habitually throughout the manuscript when referring to projections of climate change and glacial discharge regimes. These are ultimately predictions and language should reflect this.

*Reply: Thank you for this valuable comment. We agree and replaced the word "will" with more appropriate terms like "predicted" or "forecasted".*

**L70 –** there is no clear research gap or problem that has been communicated. More generally, there needs to be a hypothesis or research question that is tested. Developing this will help in linking up analysis, discussion, and conclusions.

*Reply: Thank you for this important comment. We added more detailed information about the research gap and the hypothesis of our study:*
*Proglacial rivers in a diamictic glacier foreland, a well-known dynamic system, show evidence of stabilization by glacifluvial erosion. Our aim is to verify the gradual channel bed coarsening and the tendency of pavement layer formation by hydraulic indices.*

**L76 –** reference to a 'landform decoupling'. It is not clear what this is exactly, and similar process interpretations throughout need to be explained in specific terms.

*Reply: In the initial MS (LN41), we described sediment connectivity in different directions (longitudinal, lateral, vertical) as a crucial parameter for sediment contribution for the glacifluvial transport. In turn, different landforms prevent sediment transport as they act as buffers, which decouple different (storage) landforms (Brierley et al., 2006; Fryris et al., 2007). We reworded this section and added new information during the revision.*

**L105 –** These two processes of reworking are related, however.

*Reply: We rephrased this paragraph during the revision.*

**L105-108** – Some of these statements about the dynamics of this site and potential for different processes are presented as rather factual, when they appear to be based on 1-2 studies. It may be useful to talk about these key studies and their methodologies so it's clear what has been demonstrated and how (e.g. Geilhausen et al., 2012b).

> *Reply: This section was intended to represent the study area for a more detailed overview. However, we considered this comment during the revision and have been more careful in the exact wording.*

**L139** – How was >150 stones decided? There are several rules-of-thumb across the literature, and this is not necessarily insufficient, however, some recent work has attempted to improve sampling and introduce a quantification of uncertainty. I will note one such effort by Eaton et al. 2019 that would help to demonstrate differences between GSDs more convincingly.

> *Reply: (1) We used the approach according to Fehr (1987), which is the state-of-the-art field method for gravel to cobble-bed mountain rivers. According to this method, 150 stones are required to represent the respective site adequately.*
>
> *(2) We agree with the authors of the suggested paper, that a larger number of stones result in less uncertainty but – in turn – is more time-consuming (sampling 150 stones for an LbN analysis takes around 30 minutes). However, even with the uncertainty described by the authors ($d_{50}$ varies from ±25 % for a sample size of 100 stones from a gravel bed river), our study results can still demonstrate the tendency of pavement layer formation in a proglacial diamictic environment.*

**L147** – Usage of a 1D model should be justified given there is good drone data and there are likely important lateral processes occurring here which cannot be accounted for without a 2D approach.

> *Reply: The presented study focuses on channel incision and channel bed stabilization processes. Detailed process reconstruction is also limited in a 2D modeling approach. Thus, a 1D model is sufficient and frequently applied in the glacial environment investigating glacifluvial processes (see previous reply).*

**L152** – there is no detail provided here regarding GERM, and this is needed for the study to be reproducible

> *Reply: We added more information about the GERM model in the revised MS.*

**L170** – There needs to be more detail here regarding both the orthophoto and the automatic camera. Especially with the limited temporal resolution of the dataset, the timing of these captures is critical.

> *Reply: We only used the comparison with the orthophoto of 2015 to show the year of the beginning of the channel formation. To avoid misunderstandings, shift this part to the study site description. All our results are now based on the hydrological model (GERM), the hydraulic model results, and sediment analysis.*

**L177** – narrowly graded, based on what criteria? There are indices to indicate the degree of gradation. I would be surprised if a proglacial stream was narrowly graded!

> *Reply: Thank you for this comment. We agree and rephrased this sentence.*

**L180** – to me, describing these points as 'characteristic' would imply they are representative of whatever process are of interest. They seem to have been more arbitrarily selected

> *Reply: We did the digital line sampling in all non-wetted areas in the canyon (five possible sites in total). We agree with the reviewer and replaced the term 'characteristic' in the revised MS with the more appropriate term 'specific'.*

**Figure 3** – how has the 'potential future grain size distribution' been developed?

> *Reply: It only represents a tendency and is based on the calculation results according to the highest flow competence in the year 2030.*

**L196** – What is the justification for using D50 as the characteristic grain size? In gravel-bedded streams, a larger-than-average grain size percentile is usually more appropriate as it has greater influence over bedload transport (see Mackenzie et al. 2018).

> *Reply: The approach according to Rickenmann is well-known and requires the characteristic grain size $d_{50}$ (Eq.1 in the initial MS – LN159). For considering the increased flow resistance due to macro roughness elements (like in the canyon), the energy gradient was reduced according to Eq.2 in the initial MS (LN163). For this calculation step, the $d_{90}$ is required.*

**L198-204** – It is difficult to assess the difference between these values without an estimate of uncertainty and error across the methods. Is a predicted mobile D50 diameter of 60 mm bigger than an observed surface D50 diameter of 50 mm in a way that is statistically significant? Sediment transport equations are not known for their high accuracy. Also, what does 'no big roughness elements' mean? Moreover, the assumption that the surface grain size distribution is representative of the sediment load needs to be more carefully addressed.

*Reply: (1) As described in a previous reply, the uncertainties of sediment sampling are now more appropriately addressed in the revised MS.*

*(2) Sampling surface grain sizes for analyzing armor and/or pavement layer formation is representative only to a certain extent. However, we present indications and tendencies based on the hydrological model GERM and hydraulic model data (flow competence).*

*(3) Big roughness elements like boulders or step-pool sequences create increased flow resistance in steep mountain rivers and are essential elements controlling bedload transport. The more appropriate term' macro-roughness elements' is used in literature (e.g., Nitsche et al., 2011). Therefore, we used this term in the revised MS.*

**L219** – what does 'pronounced river structure' mean?

*Reply: With this term, we described the well-visible proglacial river in the year of formation. As this paragraph describes the past morphological development, we revised the entire paragraph and transferred parts of this section to the study area description (see other replies).*

**L237-239** – What is the evidence for this? How can one be sure this will occur at this site? The authors should be careful in the discussion to be clear what evidence there is for the specific area of study, compared to studies of other areas. There also needs to be discussion of what is meant by 'channel stability', as this varies widely.

*Reply: (1) As it is cited in the initial MS, Pralong et al. (2015) observed limitations in the bedload transport triggered by altered discharge patterns. Especially in the presented study site, Avian et al. (2018) as well as Geilhausen et al. (2012), already observed less bedload transport (initial MS – LN252).*

*(2) As it was already mentioned by the reviewer and as it is stated in the initial MS (e.g., LN42, 47, 107, 220), proglacial areas are highly dynamic systems. High-magnitude/low-frequency events can increase sediment transport and great morphological changes (initial MS – LN49, 291). However, different landforms are able to interrupt sediment connectivity (e.g., Fryris et al., 2007; Brierley et al., 2006). One of these landforms is glacial deposited non-fluvial sediment forming a pavement layer and forcing landform decoupling in the vertical direction (Brierley et al., 2006; initial MS – LN284).*

*(3) The revision focused on more accurate wording, as we investigated the channel bed's gradual stabilization by the tendency of pavement layer formation. Therefore, the term 'channel stability' may lead to misunderstandings, and we used more appropriate and consistent wording in the revised MS.*

**Figure 6** – How have these 'erosion breakpoints' been defined? They appear remarkably spatially periodic; do they have physical meaning or are they just points that have selected for analysis?

*Reply: As mentioned in a previous reply, digital line sampling was applied in all non-wetted gravel bars in the canyon (five possible areas in total). According to the hydraulic model results, a tendency of pavement layer formation is already given in all five sites.*

**L296-299** – there is some attempt here to discuss the limitations of the methodology, but this deserves a more comprehensive effort to demonstrate that these limitations do not undermine the findings. Relatedly, why is it acceptable that the relatively finer fractions were not accounted for? There is a large literature on the importance of fine sediment for decreasing the entrainment threshold of larger grains. What are the limitations of only sampling surface grains, as opposed to the bulk?

*Reply: The line-by-number method according to Fehr (1987) is state-of-the-art for gravel to cobble-bed mountain rivers (see previous reply). As also mentioned by Fehr (1987) and later by Lang et al. (2021), the undersampled finer fractions are considered (or predicted) by a conversion calculation step.*

**Section 5.2 and Conclusion:** I find these sections unconvincing because they are largely unrelated to the data that has been presented. They appear to summarise general findings from the literature rather than detail the empirical and theoretical contributions of the study. The final clause 'as proven and described in this study' is inappropriate. This is, however, mostly an exercise in adjusting the conclusions to better reflect the work that's been done. I encourage them to think carefully through this process!

*Reply: We agree with the reviewer and revised the mentioned sections carefully and reflected our investigation more based on our empirical data. We also agree, that the phrase 'proven by this study' is inappropriate and we deleted it during the revision.*

**3. References**

Eaton, Brett C., R. Dan Moore, and Lucy G. MacKenzie. "Percentile-based grain size distribution analysis tools (GSDtools) – estimating confidence limits and hypothesis tests for comparing two samples." Earth Surface Dynamics 7.3 (2019): 789-806.

MacKenzie, L.G., Eaton, B.C. and Church, M., 2018. Breaking from the average: Why large grains matter in gravel-bed streams. Earth Surface Processes and Landforms, 43(15), pp.3190-3196.

Wilcock, P.R. and McArdell, B.W., 1993. Surface-based fractional transport rates: Mobilization thresholds and partial transport of a sand-gravel sediment. Water Resources Research, 29(4), pp.1297-1312.

Wilcock, P.R. and McArdell, B.W., 1997. Partial transport of a sand/gravel sediment. Water Resources Research, 33(1), pp.235-245.

*Reply: Thank you for the references. We have implemented them appropriately.*